# Mutual Information Estimation via $f$-Divergence and Data Derangements

**Nunzio A. Letizia**   **Nicola Novello**   **Andrea M. Tonello**
University of Klagenfurt
{nunzio.letizia,nicola.novello,andrea.tonello}@aau.at

## Abstract

Estimating mutual information accurately is pivotal across diverse applications, from machine learning to communications and biology, enabling us to gain insights into the inner mechanisms of complex systems. Yet, dealing with high-dimensional data presents a formidable challenge, due to its size and the presence of intricate relationships. Recently proposed neural methods employing variational lower bounds on the mutual information have gained prominence. However, these approaches suffer from either high bias or high variance, as the sample size and the structure of the loss function directly influence the training process. In this paper, we propose a novel class of discriminative mutual information estimators based on the variational representation of the $f$-divergence. We investigate the impact of the permutation function used to obtain the marginal training samples and present a novel architectural solution based on derangements. The proposed estimator is flexible since it exhibits an excellent bias/variance trade-off. The comparison with state-of-the-art neural estimators, through extensive experimentation within established reference scenarios, shows that our approach offers higher accuracy and lower complexity.

## 1 Introduction

The mutual information (MI) between two multivariate random variables, $X$ and $Y$, is a fundamental quantity in statistics, representation learning, information theory, communication engineering and biology [1, 2, 3, 4, 5]. It quantifies the statistical dependence between $X$ and $Y$ by measuring the amount of information obtained about $X$ via the observation of $Y$, and it is defined as

$$I(X;Y) = \mathbb{E}_{(\mathbf{x},\mathbf{y}) \sim p_{XY}(\mathbf{x},\mathbf{y})}\left[\log \frac{p_{XY}(\mathbf{x},\mathbf{y})}{p_X(\mathbf{x})p_Y(\mathbf{y})}\right]. \tag{1}$$

Unfortunately, computing $I(X;Y)$ is challenging since the joint probability density function $p_{XY}(\mathbf{x},\mathbf{y})$ and the marginals $p_X(\mathbf{x}), p_Y(\mathbf{y})$ are usually unknown, especially when dealing with high-dimensional data. Some recent techniques [6, 7] have demonstrated that neural networks can be leveraged as probability density function estimators and, more in general, are capable of modeling the data dependence. Discriminative approaches [8, 9] compare samples from both the joint and marginal distributions to directly compute the density ratio (or the log-density ratio)

$$R(\mathbf{x},\mathbf{y}) = \frac{p_{XY}(\mathbf{x},\mathbf{y})}{p_X(\mathbf{x})p_Y(\mathbf{y})}. \tag{2}$$

We focus on discriminative MI estimation since it can in principle enjoy some of the properties of implicit generative models, which are able of directly generating data that belongs to the same distribution of the input data without any explicit density estimate. In this direction, the most

38th Conference on Neural Information Processing Systems (NeurIPS 2024).

successful technique is represented by generative adversarial networks (GANs) [10]. The adversarial training pushes the discriminator $D(\mathbf{x})$ towards the optimum value

$$\hat{D}(\mathbf{x}) = \frac{p_{data}(\mathbf{x})}{p_{data}(\mathbf{x}) + p_{gen}(\mathbf{x})} = \frac{1}{1 + \frac{p_{gen}(\mathbf{x})}{p_{data}(\mathbf{x})}}. \tag{3}$$

Therefore, the output of the optimum discriminator is itself a function of the density ratio $p_{gen}/p_{data}$, where $p_{gen}$ and $p_{data}$ are the distributions of the generated and the collected data, respectively.

We generalize the observation of (3) and we propose a family of MI estimators based on the variational lower bound (VLB) of the $f$-divergence [11, 12]. In particular, we argue that the maximization of any $f$-divergence VLB can lead to a MI estimator with excellent bias/variance trade-off.

Since we typically have access only to joint data points $(\mathbf{x}, \mathbf{y}) \sim p_{XY}(\mathbf{x}, \mathbf{y})$, another relevant practical aspect is the sampling strategy to obtain data from the product of marginals $p_X(\mathbf{x})p_Y(\mathbf{y})$, for instance via a shuffling mechanism along $N$ realizations of $Y$. We analyze the impact that the permutation has on the learning and training process and we propose a derangement training strategy that achieves high performance requiring $\Omega(N)$ operations. Simulation results demonstrate that the proposed approach exhibits improved estimations in a multitude of scenarios.

In brief, we can summarize our contributions over the state-of-the-art as follows:

- For any $f$-divergence, we derive a training value function whose maximization leads to a given MI estimator.
- We compare different $f$-divergences and comment on the resulting estimator properties and performance.
- We study the impact of data derangement for the learning model and propose a novel derangement training strategy that overcomes the upper bound on the MI estimation [13], contrarily to what happens when using a random permutation strategy.
- We unify the main discriminative estimators into a publicly available code which can be used to reproduce all the results of this paper.

## 2 Related Work

Traditional approaches for the MI estimation rely on binning, density and kernel estimation [14, 15], $k$-nearest neighbors [16], and ensemble-based models [17]. Nevertheless, they do not scale to problems involving high-dimensional data as it is the case in modern machine learning applications. Hence, deep neural networks have recently been leveraged to maximize VLBs on the MI [11, 18, 19]. The expressive power of neural networks has shown promising results in this direction although less is known about the effectiveness of such estimators [20], especially since they suffer from either high bias or high variance.

Discriminative approaches usually exploit an energy-based variational family of functions to provide a lower bound on the Kullback-Leibler (KL) divergence. As an example, the Donsker-Varadhan dual representation of the KL divergence [11, 21] produces an estimate of the MI using the bound optimized by the mutual information neural estimator (MINE) [19]. Another VLB based on the KL divergence dual representation introduced in [18] leads to the NWJ estimator (also referred to as $f$-MINE in [19]). Both MINE and NWJ suffer from high-variance estimates and to combat such a limitation, the SMILE estimator was introduced in [20], where the authors proved that the estimate of the partition function is the cause for high-variance in VLB estimators. SMILE is equivalent to MINE in the limit $\tau \to +\infty$. The MI estimator based on contrastive predictive coding (CPC) [22] provides low variance estimates but it is upper bounded by $\log N$, resulting in a biased estimator. Such upper bound, typical of contrastive learning objectives, has been recently analyzed in the context of skew-divergence estimators [23].

Another estimator based on a classification task is the neural joint entropy estimator (NJEE) proposed in [24], which estimates the MI as entropies subtraction.

Inspired by the $f$-GAN training objective [25], in the following, we present a class of discriminative MI estimators based on the $f$-divergence measure. Conversely to what has been proposed so far in the literature, where $f$ is always constrained to be the generator of the KL divergence, we allow for

any choice of $f$. Different $f$ functions will have different impact on the training and optimization sides, while on the estimation side, the partition function does not need to be computed, leading to low variance estimators.

## 3   $f$-Divergence Mutual Information Estimation

The calculation of the MI via a discriminative approach requires the density ratio (2). From (3), we observe that $I(X; Y)$ can be estimated using the optimum GAN discriminator $\hat{D}$ when $p_{data} \equiv p_X p_Y$ and $p_{gen} \equiv p_{XY}$. More in general, the authors in [25] extended the variational divergence estimation framework presented in [18] and showed that any $f$-divergence can be used to train GANs. Inspired by such idea, we now argue that also discriminative MI estimators enjoy similar properties if the variational representation of $f$-divergence functionals $D_f(P||Q)$ is adopted.

In detail, let $P$ and $Q$ be absolutely continuous measures w.r.t. $\mathrm{d}x$ and assume they possess densities $p$ and $q$, then the $f$-divergence is defined as follows

$$D_f(P||Q) = \int_{\mathcal{X}} q(\mathbf{x}) f\left(\frac{p(\mathbf{x})}{q(\mathbf{x})}\right) \mathrm{d}\mathbf{x}, \tag{4}$$

where $\mathcal{X}$ is a compact domain and the function $f : \mathbb{R}_+ \to \mathbb{R}$ is convex, lower semicontinuous and satisfies $f(1) = 0$.

In the following, we introduce $f$-DIME, a class of discriminative mutual information estimators (DIME) based on the variational representation of the $f$-divergence.

**Theorem 3.1.** *Let $(X, Y) \sim p_{XY}(\mathbf{x}, \mathbf{y})$ be a pair of multivariate random variables. Let $\sigma(\cdot)$ be a permutation function such that $p_{\sigma(Y)}(\sigma(\mathbf{y})|\mathbf{x}) = p_Y(\mathbf{y})$ and $T : \mathrm{dom}(X) \times \mathrm{dom}(Y) \to \mathbb{R}$. Let $f^*$ be the Fenchel conjugate of $f : \mathbb{R}_+ \to \mathbb{R}$, a convex lower semicontinuous function that satisfies $f(1) = 0$ with derivative $f'$. If $\mathcal{J}_f(T)$ is a value function defined as*

$$\mathcal{J}_f(T) = \mathbb{E}_{(\mathbf{x}, \mathbf{y}) \sim p_{XY}(\mathbf{x}, \mathbf{y})}\Big[T(\mathbf{x}, \mathbf{y}) - f^*\Big(T(\mathbf{x}, \sigma(\mathbf{y}))\Big)\Big], \tag{5}$$

*then*

$$\hat{T}(\mathbf{x}, \mathbf{y}) = \arg\max_T \mathcal{J}_f(T) = f'\left(\frac{p_{XY}(\mathbf{x}, \mathbf{y})}{p_X(\mathbf{x}) p_Y(\mathbf{y})}\right), \tag{6}$$

*and*

$$I(X; Y) = I_{fDIME}(X; Y) = \mathbb{E}_{(\mathbf{x}, \mathbf{y}) \sim p_{XY}(\mathbf{x}, \mathbf{y})}\Big[\log\Big((f^*)'(\hat{T}(\mathbf{x}, \mathbf{y}))\Big)\Big]. \tag{7}$$

Theorem 3.1 shows that any value function $\mathcal{J}_f$ of the form in (5), seen as the dual representation of a given $f$-divergence $D_f$, can be maximized to estimate the MI via (7). It is interesting to notice that the proposed class of estimators does not need any evaluation of the partition term.

We propose to parametrize $T(\mathbf{x}, \mathbf{y})$ with a deep neural network $T_\theta$ of parameters $\theta$ and solve with gradient ascent and back-propagation to obtain $\hat{\theta} = \arg\max_\theta \mathcal{J}_f(T_\theta)$. By doing so, it is possible to guarantee that, at every training iteration $n$, the convergence of the $f$-DIME estimator $\hat{I}_{n,fDIME}(X; Y)$ is controlled by the convergence of $T$ towards the tight bound $\hat{T}$ while maximizing $\mathcal{J}_f(T)$, as stated in the following lemma.

**Lemma 3.2.** *Let the discriminator $T(\cdot)$ be with enough capacity, i.e., in the non parametric limit. Consider the problem*

$$\hat{T} = \arg\max_T \mathcal{J}_f(T) \tag{8}$$

*where $\mathcal{J}_f(T)$ is defined as in (5), and the update rule based on the gradient descent method*

$$T^{(n+1)} = T^{(n)} + \mu \nabla \mathcal{J}_f(T^{(n)}). \tag{9}$$

*If the gradient descent method converges to the global optimum $\hat{T}$, the mutual information estimator defined in (7) converges to the real value of the mutual information $I(X; Y)$.*

The proof of Lemma 3.2, which is described in the Appendix, provides some theoretical grounding for the behaviour of MI estimators when the training does not converge to the optimal density ratio. Moreover, it also offers insights about the impact of different functions $f$ on the numerical bias.

It is important to remark the difference between the classical VLB estimators that follow a discriminative approach and the DIME-like estimators. They both achieve the goal through a discriminator network that outputs a function of the density ratio. However, the former models exploit the variational representation of the MI (or the KL) and, at the equilibrium, use the discriminator output directly in one of the value functions reported in Appendix B. The latter, instead, use the variational representation of *any* $f$-divergence to extract the density ratio estimate directly from the discriminator output.

In the upcoming sections, we analyze the variance of $f$-DIME and we propose a training strategy for the implementation of Theorem 3.1. In our experiments, we consider the cases when $f$ is the generator of: a) the KL divergence; b) the GAN divergence; c) the Hellinger distance squared. Due to space constraints, we report in Sec. A of the Appendix the value functions used for training and the mathematical expressions of the resulting DIME estimators.

## 4 Variance Analysis

In this section, we assume that the ground truth density ratio $\hat{R}(\mathbf{x}, \mathbf{y})$ exists and corresponds to the density ratio in (2). We also assume that the optimum discriminator $\hat{T}(\mathbf{x}, \mathbf{y})$ is known and already obtained (e.g. via a neural network parametrization).

We define $p_{XY}^M(\mathbf{x}, \mathbf{y})$ and $p_X^N(\mathbf{x}) p_Y^N(\mathbf{y})$ as the empirical distributions corresponding to $M$ i.i.d. samples from the true joint distribution $p_{XY}$ and to $N$ i.i.d. samples from the product of marginals $p_X p_Y$, respectively. The randomness of the sampling procedure and the batch sizes $M, N$ influence the variance of variational MI estimators. In the following, we prove that under the previous assumptions, $f$-DIME exhibits better performance in terms of variance w.r.t. some variational estimators with a discriminative approach, e.g., MINE and NWJ.

The partition function estimation $\mathbb{E}_{p_X^N p_Y^N}[\hat{R}]$ represents the major issue when dealing with variational MI estimators. Indeed, they comprise the evaluation of two terms (using the given density ratio), and the partition function is the one responsible for the variance growth. The authors in [20] characterized the variance of both MINE and NWJ estimators, in particular, they proved that the variance scales exponentially with the ground truth MI $\forall M \in \mathbb{N}$

$$\text{Var}_{p_{XY}, p_X p_Y}\left[I_{NWJ}^{M,N}\right] \geq \frac{e^{I(X;Y)} - 1}{N}$$
$$\lim_{N \to \infty} N \text{Var}_{p_{XY}, p_X p_Y}\left[I_{MINE}^{M,N}\right] \geq e^{I(X;Y)} - 1, \tag{10}$$

where

$$I_{NWJ}^{M,N} := \mathbb{E}_{p_{XY}^M}[\log \hat{R} + 1] - \mathbb{E}_{p_X^N p_Y^N}[\hat{R}]$$
$$I_{MINE}^{M,N} := \mathbb{E}_{p_{XY}^M}[\log \hat{R}] - \log \mathbb{E}_{p_X^N p_Y^N}[\hat{R}]. \tag{11}$$

To reduce the impact of the partition function on the variance, the authors of [20] also proposed to clip the density ratio between $e^{-\tau}$ and $e^{\tau}$ leading to an estimator (SMILE) with bounded partition variance. However, also the variance of the log-density ratio $\mathbb{E}_{p_{XY}^M}[\log \hat{R}]$ influences the variance of the variational estimators, since it is clear that

$$\text{Var}_{p_{XY}, p_X p_Y}\left[I_{VLB}^{M,N}\right] \geq \text{Var}_{p_{XY}}\left[\mathbb{E}_{p_{XY}^M}[\log \hat{R}]\right], \tag{12}$$

a result that holds for any type of MI estimator based on a VLB.

The great advantage of $f$-DIME is to avoid the partition function estimation step, significantly reducing the variance of the estimator. Under the same initial assumptions, from (12) we can immediately conclude that

$$\text{Var}_{p_{XY}}\left[I_{fDIME}^M\right] \leq \text{Var}_{p_{XY}, p_X p_Y}\left[I_{VLB}^{M,N}\right], \tag{13}$$

where

$$I_{fDIME}^M := \mathbb{E}_{p_{XY}^M}[\log \hat{R}] \tag{14}$$

is the Monte Carlo implementation of $f$-DIME. Hence, the $f$-DIME class of models has lower variance than any VLB based estimator (MINE, NWJ, SMILE, etc.).

Furthermore, we provide in Appendix C two supplementary results. Lemma 4.1 introduces an upper bound on the variance of the $f$-DIME estimator, a result holding for any type of value function $\mathcal{J}_f$. Lemma 4.2, instead, characterizes the variance of the estimator in (14) when $X$ and $Y$ are correlated Gaussian random variables. We found out that the variance is finite and we use this result to verify in the experiments that the variance of $f$-DIME does not diverge for high values of MI.

## 5  Derangement Strategy

The discriminative approach essentially compares expectations over both joint $(\mathbf{x}, \mathbf{y}) \sim p_{XY}$ and marginal $(\mathbf{x}, \mathbf{y}) \sim p_X p_Y$ data points. Practically, we have access only to $N$ realizations of the joint distribution $p_{XY}$ and to obtain $N$ marginal samples of $p_X p_Y$ from $p_{XY}$ a shuffling mechanism for the realizations of $Y$ is typically deployed. A general result in [13] shows that failing to sample from the correct marginal distribution would lead to an upper bounded MI estimator.

We study the structure that the permutation law $\sigma(\cdot)$ in Theorem 3.1 needs to have when numerically implemented. In particular, we now prove that a naive permutation over the realizations of $Y$ results in an incorrect VLB of the $f$-divergence, causing the MI estimator to be bounded by $\log(N)$, where $N$ is the batch size. To solve this issue, we propose a derangement strategy.

Let the data points $(\mathbf{x}, \mathbf{y}) \sim p_{XY}$ be $N$ pairs $(\mathbf{x}_i, \mathbf{y}_i)$, $\forall i \in \{1, \ldots, N\}$. The naive permutation of $\mathbf{y}$, denoted as $\pi(\mathbf{y})$, leads to $N$ new random pairs $(\mathbf{x}_i, \mathbf{y}_j)$, $\forall i$ and $j \in \{1, \cdots, N\}$. The idea is that a random naive permutation may lead to at least one pair $(\mathbf{x}_k, \mathbf{y}_k)$, with $k \in \{1, \ldots, N\}$, which is actually a sample from the joint distribution. Viceversa, the derangement of $\mathbf{y}$, denoted as $\sigma(\mathbf{y})$, leads to $N$ new random pairs $(\mathbf{x}_i, \mathbf{y}_j)$ such that $i \neq j, \forall i$ and $j \in \{1, \cdots, N\}$. Such pairs $(\mathbf{x}_i, \mathbf{y}_j), i \neq j$ can effectively be considered samples from $p_X(\mathbf{x}) p_Y(\mathbf{y})$. An example using these definitions is provided in Appendix D.1.3.

The following lemma analyzes the relationship between the Monte Carlo approximations of the VLBs of the $f$-divergence $\mathcal{J}_f$ in Theorem 3.1 using $\pi(\cdot)$ and $\sigma(\cdot)$ as permutation laws.

**Lemma 5.1.** *Let $(\mathbf{x}_i, \mathbf{y}_i)$, $\forall i \in \{1, \ldots, N\}$, be $N$ data points. Let $\mathcal{J}_f(T)$ be the value function in (5). Let $\mathcal{J}_f^\pi(T)$ and $\mathcal{J}_f^\sigma(T)$ be numerical implementations of $\mathcal{J}_f(T)$ using a random permutation and a random derangement of $\mathbf{y}$, respectively. Denote with $K$ the number of points $\mathbf{y}_k$, with $k \in \{1, \ldots, N\}$, in the same position after the permutation (i.e., the fixed points). Then*

$$\mathcal{J}_f^\pi(T) \leq \frac{N-K}{N} \mathcal{J}_f^\sigma(T). \tag{15}$$

Lemma 5.1 practically asserts that the value function $\mathcal{J}_f^\pi(T)$ evaluated via a naive permutation of the data is not a valid VLB of the $f$-divergence, and thus, there is no guarantee on the optimality of the discriminator's output. An interesting mathematical connection can be obtained when studying $\mathcal{J}_f^\pi(T)$ as a sort of variational skew-divergence estimator [23], but this goes beyond the scope of this paper.

The following theorem states that in the case of the KL divergence, the maximum of $\mathcal{J}_f^\pi(D)$ is attained for a value of the discriminator that is not exactly the density ratio (as it should be from (21), see Appendix A).

**Theorem 5.2.** *Let the discriminator $D(\cdot)$ be with enough capacity. Let $N$ be the batch size and $f$ be the generator of the KL divergence. Let $\mathcal{J}_{KL}^\pi(D)$ be defined as*

$$\mathcal{J}_{KL}^\pi(D) = \mathbb{E}_{(\mathbf{x}, \mathbf{y}) \sim p_{XY}(\mathbf{x}, \mathbf{y})}\left[\log\Big(D(\mathbf{x}, \mathbf{y})\Big) - f^*\Big(\log\Big(D(\mathbf{x}, \pi(\mathbf{y}))\Big)\Big)\right]. \tag{16}$$

*Denote with $K$ the number of indices in the same position after the permutation (i.e., the fixed points), and with $R(\mathbf{x}, \mathbf{y})$ the density ratio in (2). Then,*

$$\hat{D}(\mathbf{x}, \mathbf{y}) = \arg\max_D \mathcal{J}_{KL}^\pi(D) = \frac{N R(\mathbf{x}, \mathbf{y})}{K R(\mathbf{x}, \mathbf{y}) + N - K}. \tag{17}$$

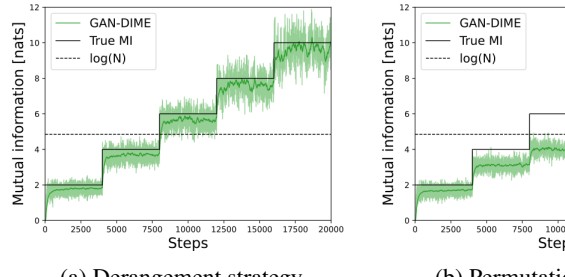

|                                | (a) Derangement strategy. | (b) Permutation strategy. |
|--------------------------------|---------------------------|---------------------------|

Figure 1: MI estimate obtained with derangement and permutation training procedures, for data dimension $d = 20$ and batch size $N = 128$.

Although Theorem 5.2 is stated for the KL divergence, it can be easily extended to any $f$-divergence using Theorem 3.1. Notice that if the number of indices in the same position $K$ is equal to 0, we fall back into the derangement strategy and we retrieve the density ratio as output.

When we parametrize $D$ with a neural network, we perform multiple training iterations and so we have multiple batches of dimension $N$. This turns into an average analysis on $K$. We report in the Appendix (see Lemma 5.4) the proof that, on average, $K$ is equal to 1.

From the previous results, it follows immediately that the estimator obtained using a naive permutation strategy is biased and upper bounded by a function of the batch size $N$.

**Corollary 5.3** (Permutation bound). *Let KL-DIME be the estimator obtained via iterative optimization of $\mathcal{J}_{KL}^{\pi}(D)$, using a batch of size $N$ every training step. Then,*

$$I_{KL-DIME}^{\pi} := \mathbb{E}_{(\mathbf{x},\mathbf{y}) \sim p_{XY}(\mathbf{x},\mathbf{y})} \left[ \log \left( \hat{D}(\mathbf{x},\mathbf{y}) \right) \right] < \log(N). \tag{18}$$

We report in Fig. 1 an example of the difference between the derangement and permutation strategies. The estimate attained by using the permutation mechanism, showed in Fig. 1b, demonstrates Theorem 5.2 and Corollary 5.3, as the upper bound corresponding to $\log(N)$ (with $N = 128$) is clearly visible.

## 6 Experimental Results

In this section, we firstly describe the architectures of the proposed estimators. Then, we outline the data used to estimate the MI, comment on the performance of the discussed estimators in different scenarios, also analyzing their computational complexity. Finally, we present the outcomes of the self-consistency tests [20] over image datasets.

### 6.1 Architectures

To demonstrate the behavior of the state-of-the-art MI estimators, we consider multiple neural network *architectures*. The word architecture needs to be intended in a wide-sense, meaning that it represents the neural network architecture and its training strategy. In particular, additionally to the architectures **joint** [19] and **separable** [26], we propose the architecture **deranged**.
The **joint** architecture concatenates the samples $\mathbf{x}$ and $\mathbf{y}$ as input of a single neural network. Each training step requires $N$ realizations $(\mathbf{x}_i, \mathbf{y}_i)$ drawn from $p_{XY}(\mathbf{x}, \mathbf{y})$, for $i \in \{1, \ldots, N\}$ and $N(N-1)$ samples $(\mathbf{x}_i, \mathbf{y}_j), \forall i, j \in \{1, \ldots, N\}$, with $i \neq j$.
The **separable** architecture comprises two neural networks, the former fed in with $N$ realizations of $X$, the latter with $N$ realizations of $Y$. The inner product between the outputs of the two networks is exploited to obtain the MI estimate.
The proposed **deranged** architecture feeds a neural network with the concatenation of the samples $\mathbf{x}$ and $\mathbf{y}$, similarly to the joint architecture. However, the deranged one obtains the samples of $p_X(\mathbf{x})p_Y(\mathbf{y})$ by performing a derangement of the realizations $\mathbf{y}$ in the batch sampled from $p_{XY}(\mathbf{x}, \mathbf{y})$. Such diverse training strategy solves the main problem of the joint architecture: the difficult scalability to large batch sizes. For large values of $N$, the complexity of the joint architecture is $\Omega(N^2)$, while

the complexity of the deranged one is $\Omega(N)$. NJEE utilizes a specific architecture, in the following referred to as **ad hoc**, comprising $2d - 1$ neural networks, where $d$ is the dimension of $X$. $I_{NJEE}$ training procedure is supervised: the input of each neural network does not include the $\mathbf{y}$ samples. All the implementation details[1] are reported in Appendix D.

## 6.2 Complex Gaussian and non-Gaussian distributions

We benchmark the proposed class of MI estimators on two settings utilized in previous papers [11, 20]. In the first setting (called **Gaussian**), a multidimensional Gaussian distribution is sampled to obtain $\mathbf{x}$ and $\mathbf{n}$ samples, independently. Then, $\mathbf{y}$ is obtained as linear combination of $\mathbf{x}$ and $\mathbf{n}$: $\mathbf{y} = \rho\mathbf{x} + \sqrt{1 - \rho^2}\,\mathbf{n}$, where $\rho$ is the correlation coefficient. In the second setting (referred to as **cubic**), the nonlinear transformation $\mathbf{y} \mapsto \mathbf{y}^3$ is applied to the Gaussian samples. The true MI follows a staircase shape, where each step is a multiple of $2\ nats$. Each neural network is trained for 4k iterations for each stair step, with a batch size of 64 samples ($N = 64$). The tested estimators are: $I_{NJEE}$, $I_{SMILE}$ ($\tau = 1$), $I_{GAN-DIME}$, $I_{HD-DIME}$, $I_{KL-DIME}$, and $I_{CPC}$, as illustrated in Fig. 2. The performance of $I_{MINE}$, $I_{NWJ}$, and $I_{SMILE}(\tau = \infty)$ is reported in Sec. D of the Appendix, since they exhibit lower performance compared to both SMILE and $f$-DIME. In fact, all the $f$-DIME estimators have lower variance compared to $I_{MINE}$, $I_{NWJ}$, and $I_{SMILE}(\tau = \infty)$, which are characterized by an exponentially increasing variance (see (10), Tab. 2, Fig. 9, and Fig. 6 in the Appendix). In particular, all the estimators analyzed belonging to the $f$-DIME class achieve significantly low bias and variance when the true MI is small. Interestingly, for high target MI, different $f$-divergences lead to dissimilar estimation properties. For large MI, $I_{KL-DIME}$ is characterized by a low variance, at the expense of a high bias and a slow rise time. Contrarily, $I_{HD-DIME}$ attains a lower bias at the cost of slightly higher variance w.r.t. $I_{KL-DIME}$. Diversely, $I_{GAN-DIME}$ achieves the lowest bias, and a variance comparable to $I_{HD-DIME}$. Additional results confirming the estimators' behavior when $d$ and $N$ vary, including experiments with high data dimensionality, are reported and described in Appendix D.

$I_{NJEE}$ obtains an estimate which is highly biased, and variance comparable to $f$-DIME. $I_{CPC}$ is upper-bounded by $\log(N)$. The MI estimates obtained with $I_{SMILE}$ and $I_{GAN-DIME}$ appear to possess similar behavior, although the value functions of SMILE and GAN-DIME are structurally different. The reason why $I_{SMILE}$ is almost equivalent to $I_{GAN-DIME}$ resides in their training strategy, since they both minimize the same $f$-divergence. Looking at the implementation of SMILE [2], in fact, the network's training is guided by the gradient computed using the Jensen-Shannon (JS) divergence (a linear transformation of the GAN divergence). Given the trained network, the clipped objective function proposed in [20] is only used to compute the MI estimate, since when (29) is used to train the network, the MI estimate diverges (see Fig. 7 in Appendix D). However, with the proposed class of $f$-DIME estimators we show that during the estimation phase the partition function (clipped in [20]) is not necessary to obtain the MI estimate.

We test our estimators over additional complex Gaussian data transformations (half-cube, asinh, and swiss roll mappings, Fig. 3) and non-Gaussian distributions (uniform and student distributions, Fig. 4) as suggested in [27]. The half-cube mapping is used to lengthen the tails of the Gaussian distributions. The inverse hyperbolic sine (asinh) mapping shortens the tails of the Gaussian distributions. These two transformations are applied to the same scenario of the Gaussian and cubic already present in our paper. The swiss roll mapping increases the dimensionality of the data distribution (from two to three dimensions) and it is usually used to test dimensionality reduction techniques. It considers two Gaussian random variables that are transformed into uniform random variables via the probability integral transform, the same pre-processing approach utilized in [28] to estimate the MI. The swiss roll mapping is applied to the $X$ uniform random variable. The stairs plots are obtained by varying the correlation between the initial Gaussian distributions. The uniform case estimates the MI of the summation of two uniform random variables $U(0, 1)$ and $U(-\epsilon, \epsilon)$, where we vary the parameter $\epsilon$, modifying the true MI. The student scenario analyzes the case of a multivariate student distribution with dispersion matrix chosen to be the identity matrix and degrees of freedom $df$. In this scenario, we vary $df$, implying a variation of the target MI. For the transformed Gaussian scenarios showed in Fig. 3 GAN-DIME attains the best performance in terms of low bias and variance. Among the

---

[1] Our implementation can be found at `https://github.com/tonellolab/fDIME`

[2] `https://github.com/ermongroup/smile-mi-estimator`

non-Gaussian settings depicted in Fig. 4, KL-DIME and GAN-DIME outperform the other methods, exhibiting low bias and exceptionally low variance.

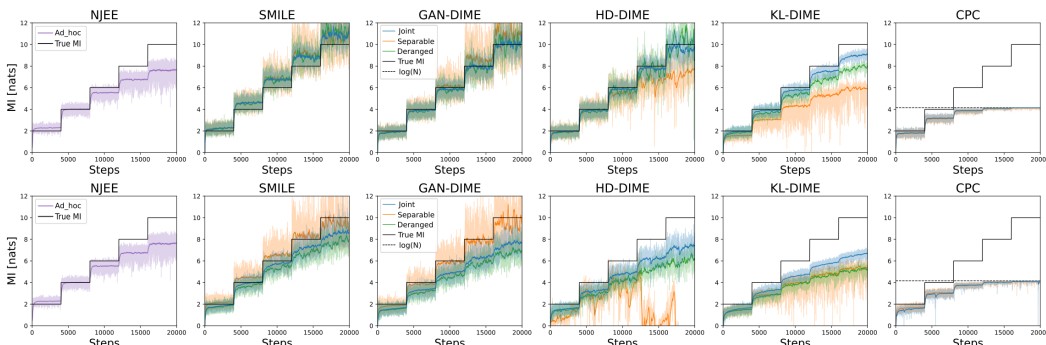

Figure 2: Staircase MI estimation comparison for $d = 5$ and $N = 64$. The *Gaussian* case is reported in the top row, while the *cubic* case is shown in the bottom row.

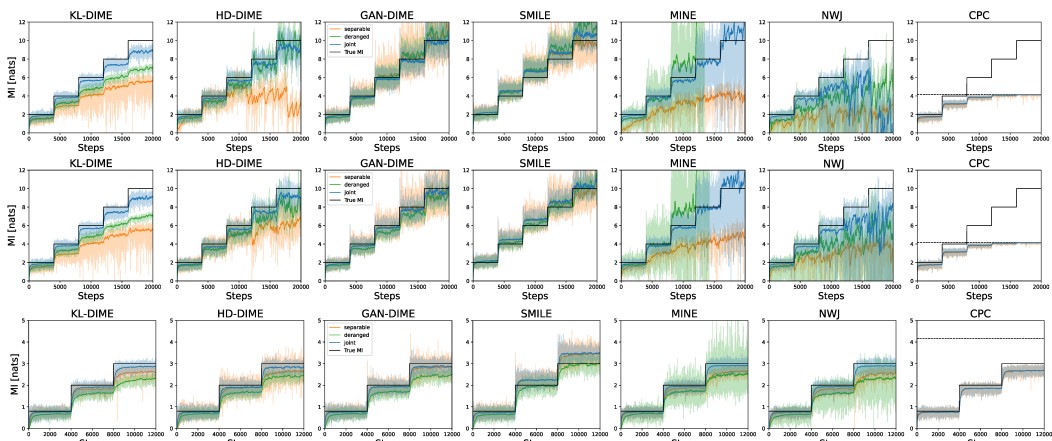

Figure 3: Staircase MI estimation comparison for $d = 5$ and $N = 64$. Top: Half-cube scenario. Middle: Asinh scenario. Bottom: Swiss roll scenario.

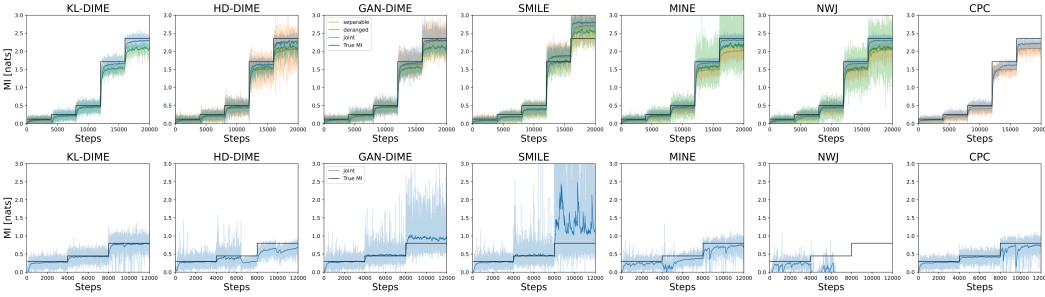

Figure 4: Staircase MI estimation comparison for $d = 1$ and $N = 64$. Top row: Uniform scenario. Bottom row: Student scenario

A schematic comparison between all the MI estimators is reported in Tab. 6 in Sec. D of the Appendix, where $I_{GAN-DIME}$ is proposed as the best estimator, because of its low bias, variance and robustness to the change of $d$ and $N$. When $N$ and $d$ vary, in fact, the class of $f$-DIME estimators proves its robustness (i.e., maintains low bias and variance), as represented in Figs. 2, and 10, and 11 in the Appendix. For instance, $I_{GAN-DIME}$ attains low bias in all the three scenarios, and limited variance which decreases as $N$ increases (see also Fig. 15 in Appendix D.1). Differently, the behavior

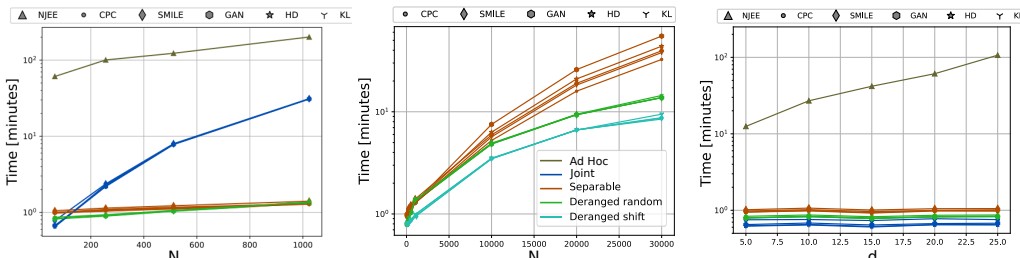

Figure 5: Time requirements comparison to complete the 5-step staircase MI. From the left, the first and second behaviors vary over the batch size. The last one varies over the probability distribution dimension.

of $I_{CPC}$ strongly depends on $N$, significantly impacting its bias. Therefore, unless the batch size is considerably large, $I_{CPC}$ estimate is not reliable. $I_{NJEE}$ attains higher bias when $N$ increases and, even more severely, when $d$ decreases (see Fig. 2).

**Computational Time Analysis**

A fundamental characteristic of each algorithm is the computational time. The computational time analysis is developed on a server with CPU "AMD Ryzen Threadripper 3960X 24-Core Processor" and GPU "MSI GeForce RTX 3090 Gaming X Trio 24G, 24GB GDDR6X".

Before analyzing the time requirements to complete the 5-step MI staircases, we specify two different ways to implement the derangement of the $\mathbf{y}$ realizations in each batch.

**Random-based**. The trivial way to achieve the derangement is to randomly shuffle the $\mathbf{y}$ elements of the batch until there are no fixed points (i.e., all the $\mathbf{y}$ realizations in the batch are assigned to a different position w.r.t. the starting location).

**Shift-based**. Given $N$ realizations $(\mathbf{x}_i, \mathbf{y}_i)$ drawn from $p_{XY}(\mathbf{x}, \mathbf{y})$, for $i \in \{1, \ldots, N\}$, we obtain the deranged samples as $(\mathbf{x}_i, \mathbf{y}_{(i+1)\%N})$, where "%" is the modulo operator.

Although the MI estimates obtained by the two derangement methods are almost indistinguishable, all the results showed in the paper are achieved by using the random-based method. Additionally, we demonstrate the time efficiency of the shift-based approach.

The time requirements to complete the 5-step staircase MI when varying the batch size $N$ are reported in the left and center graphics of Fig. 5. The influence of the MI estimator objective functions in the algorithm's time requirements is marginal, while the architecture type is the impactful component. As discussed in Sec. 6.1, the deranged strategy is remarkably faster than the joint one as $N$ increases. More in general, the architectures deranged and separable are significantly faster w.r.t. the joint and NJEE ones, for a given batch size $N$ and input distribution size $d$. The need of the separable architecture to use two neural networks implies that when $N$ is significantly large, the deranged implementation is much faster than the separable one. The central graph in Fig. 5 illustrates a detailed representation of the time requirements of these two architectures to complete the 5-step stairs. As $N$ increases, the gap between the time needed by the architectures deranged and separable grows, demonstrating that the former is the fastest. For example, when $d = 20$ and $N = 30k$, $I_{GAN-DIME}$ needs about 55 minutes when using the architecture separable, but only 15 minutes when using the deranged one and less than 9 minutes for the shift-based deranged architecture.

$I_{NJEE}$ is evaluated with its own architecture, which is the most computationally demanding, because it trains a number of neural networks equal to $2d - 1$. Thus, $I_{NJEE}$ can be utilized only in cases where the time availability is orders of magnitude higher than the other approaches considered. The time requirements to complete the 5-step staircase MI when varying the multivariate Gaussian distribution dimension $d$ are reported in the right-side part of Fig. 5. When $d$ is large, the training of $I_{NJEE}$ fails due to memory requirement problems. For example, our hardware platform does not allow the usage of $d > 30$.

### 6.3 Self-Consistency Tests

To demonstrate the utility of $f$-DIME in non-Gaussian scenarios, we investigated the three self-consistency tests developed by [20] over images datasets using all the estimators previously described, except $I_{NJEE}$ (for dimension constraints). The $f$-DIME estimators satisfy two out of the three tests,

as discriminative approaches tend to be less precise when the MI is high, in accordance with [20]. We report the description of tests and results in Appendix D.

## 7 Conclusions

In this paper, we presented $f$-DIME, a class of discriminative mutual information estimators based on the variational representation of the $f$-divergence. We proved that any valid choice of the function $f$ leads to a low-variance MI estimator which can be parametrized by a neural network. We also proposed a derangement training strategy that efficiently samples from the product of marginal distributions. The performance of $f$-DIME is evaluated using three functions $f$, and it is compared with state-of-the-art estimators. Results demonstrate excellent bias/variance trade-off for different data dimensions and different training parameters.

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

# A  Appendix: DIME Estimators

In this section, we provide a concrete list of DIME estimators obtained using three different $f$-divergences. In particular, we maximize the value function defined in (5)

$$\mathcal{J}_f(T) = \mathbb{E}_{(\mathbf{x},\mathbf{y})\sim p_{XY}(\mathbf{x},\mathbf{y})}\left[T(\mathbf{x},\mathbf{y}) - f^*\left(T(\mathbf{x},\sigma(\mathbf{y}))\right)\right],$$

over $T$ or its transformation. By doing that, and using (7),

$$I(X;Y) = I_{fDIME}(X;Y) = \mathbb{E}_{(\mathbf{x},\mathbf{y})\sim p_{XY}(\mathbf{x},\mathbf{y})}\left[\log\left((f^*)'(\hat{T}(\mathbf{x},\mathbf{y}))\right)\right],$$

we obtain a list of three different MI estimators. The list is used for both commenting on the impact of the $f$ function, referred to as the generator function, and for comparing the estimators discussed in Sec. 2.

We consider the cases when $f$ is the generator of:

a) the KL divergence;

b) the GAN divergence;

c) the Hellinger distance squared.

We report below the derived value functions and the mathematical expressions of the proposed estimators.

## A.1  KL divergence

The variational representation of the KL divergence [18] leads to the NWJ estimator in (28) when $f(u) = u\log(u)$. However, since we are interested in extracting the density ratio, we apply the transformation $T(\mathbf{x}) = \log(D(\mathbf{x}))$. In this way, the lower bound introduced in (5) reads as follows

$$\mathcal{J}_{KL}(D) = \mathbb{E}_{(\mathbf{x},\mathbf{y})\sim p_{XY}(\mathbf{x},\mathbf{y})}\left[\log\left(D(\mathbf{x},\mathbf{y})\right)\right] - \mathbb{E}_{(\mathbf{x},\mathbf{y})\sim p_X(\mathbf{x})p_Y(\mathbf{y})}\left[D(\mathbf{x},\mathbf{y})\right] + 1, \quad (19)$$

which has to be maximized over positive discriminators $D(\cdot)$. As remarked before, we do not use $\mathcal{J}_{KL}$ during the estimation, rather we define the KL-DIME estimator as

$$I_{KL-DIME}(X;Y) := \mathbb{E}_{(\mathbf{x},\mathbf{y})\sim p_{XY}(\mathbf{x},\mathbf{y})}\left[\log\left(\hat{D}(\mathbf{x},\mathbf{y})\right)\right], \quad (20)$$

due to the fact that

$$\hat{D}(\mathbf{x},\mathbf{y}) = \arg\max_D \mathcal{J}_{KL}(D) = \frac{p_{XY}(\mathbf{x},\mathbf{y})}{p_X(\mathbf{x})p_Y(\mathbf{y})}. \quad (21)$$

## A.2  GAN divergence

Following a similar approach, it is possible to define $f(u) = u\log u - (u+1)\log(u+1) + \log 4$ and $T(\mathbf{x}) = \log(1 - D(\mathbf{x}))$. We derive from Theorem 3.1 the GAN-DIME estimator obtained via maximization of

$$\mathcal{J}_{GAN}(D) = \mathbb{E}_{(\mathbf{x},\mathbf{y})\sim p_{XY}(\mathbf{x},\mathbf{y})}\left[\log\left(1 - D(\mathbf{x},\mathbf{y})\right)\right] + \mathbb{E}_{(\mathbf{x},\mathbf{y})\sim p_X(\mathbf{x})p_Y(\mathbf{y})}\left[\log\left(D(\mathbf{x},\mathbf{y})\right)\right] + \log(4).$$

$$(22)$$

In fact, at the equilibrium we recover (3), hence

$$I_{GAN-DIME}(X;Y) := \mathbb{E}_{(\mathbf{x},\mathbf{y})\sim p_{XY}(\mathbf{x},\mathbf{y})}\left[\log\left(\frac{1 - \hat{D}(\mathbf{x},\mathbf{y})}{\hat{D}(\mathbf{x},\mathbf{y})}\right)\right]. \quad (23)$$

### A.3 Hellinger distance

The last example we consider is the generator of the Hellinger distance squared $f(u) = (\sqrt{u} - 1)^2$ with the change of variable $T(\mathbf{x}) = 1 - D(\mathbf{x})$. After simple manipulations, we obtain the associated value function as

$$\mathcal{J}_{HD}(D) = 2 - \mathbb{E}_{(\mathbf{x},\mathbf{y}) \sim p_{XY}(\mathbf{x},\mathbf{y})}\left[D(\mathbf{x},\mathbf{y})\right] - \mathbb{E}_{(\mathbf{x},\mathbf{y}) \sim p_X(\mathbf{x})p_Y(\mathbf{y})}\left[\frac{1}{D(\mathbf{x},\mathbf{y})}\right], \qquad (24)$$

which is maximized for

$$\hat{D}(\mathbf{x},\mathbf{y}) = \arg\max_D \mathcal{J}_{HD}(D) = \sqrt{\frac{p_X(\mathbf{x})p_Y(\mathbf{y})}{p_{XY}(\mathbf{x},\mathbf{y})}}, \qquad (25)$$

leading to the HD-DIME estimator

$$I_{HD-DIME}(X;Y) := \mathbb{E}_{(\mathbf{x},\mathbf{y}) \sim p_{XY}(\mathbf{x},\mathbf{y})}\left[\log\left(\frac{1}{\hat{D}^2(\mathbf{x},\mathbf{y})}\right)\right]. \qquad (26)$$

Given that these estimators comprise only one expectation over the joint samples, their variance has different properties compared to the variational ones requiring the partition term such as MINE and NWJ.

## B  Appendix: Related Work Mutual Information Estimators

In this section, we provide a detailed description of the formulas of the MI estimators we summarized in Sec. 2.

### B.1  MINE

The Donsker-Varadhan dual representation of the KL divergence [11, 21] produces an estimate of the MI using the bound optimized by the mutual information neural estimator (MINE) [19]

$$I_{MINE}(X;Y) = \sup_{\theta \in \Theta} \mathbb{E}_{(\mathbf{x},\mathbf{y}) \sim p_{XY}(\mathbf{x},\mathbf{y})}[T_\theta(\mathbf{x},\mathbf{y})] - \log(\mathbb{E}_{(\mathbf{x},\mathbf{y}) \sim p_X(\mathbf{x})p_Y(\mathbf{y})}[e^{T_\theta(\mathbf{x},\mathbf{y})}]), \qquad (27)$$

where $\theta \in \Theta$ parameterizes a family of functions $T_\theta : \mathcal{X} \times \mathcal{Y} \to \mathbb{R}$ through the use of a deep neural network. However, the logarithm before the expectation in the second term renders MINE a biased estimator. To avoid biased gradients, the authors in [19] suggested to replace the partition function $\mathbb{E}_{p_X p_Y}[e^{T_\theta}]$ with an exponential moving average over mini-data-batches.

### B.2  NWJ

Another VLB is based on the KL divergence dual representation introduced in [18] (also referred to as $f$-MINE in [19])

$$I_{NWJ}(X;Y) = \sup_{\theta \in \Theta} \mathbb{E}_{(\mathbf{x},\mathbf{y}) \sim p_{XY}(\mathbf{x},\mathbf{y})}[T_\theta(\mathbf{x},\mathbf{y})] - \mathbb{E}_{(\mathbf{x},\mathbf{y}) \sim p_X(\mathbf{x})p_Y(\mathbf{y})}[e^{T_\theta(\mathbf{x},\mathbf{y})-1}]. \qquad (28)$$

Although for a fixed $T$ MINE provides a tighter bound $I_{MINE} \geq I_{NWJ}$, the NWJ estimator is unbiased.

### B.3  SMILE

Both MINE and NWJ suffer from high-variance estimations and to combat such a limitation, the SMILE estimator was introduced in [20]. It is defined as

$$I_{SMILE}(X;Y) = \sup_{\theta \in \Theta} \mathbb{E}_{(\mathbf{x},\mathbf{y}) \sim p_{XY}(\mathbf{x},\mathbf{y})}[T_\theta(\mathbf{x},\mathbf{y})] - \log(\mathbb{E}_{(\mathbf{x},\mathbf{y}) \sim p_X(\mathbf{x})p_Y(\mathbf{y})}[\text{clip}(e^{T_\theta(\mathbf{x},\mathbf{y})}, e^{-\tau}, e^{\tau})]),$$

$$(29)$$

where $\text{clip}(v, l, u) = \max(\min(v, u), l)$ and it helps to obtain smoother partition functions estimates. SMILE is equivalent to MINE in the limit $\tau \to +\infty$.

## B.4 CPC

The MI estimator based on contrastive predictive coding (CPC) [22] is defined as

$$I_{CPC}(X;Y) = \mathbb{E}_{(\mathbf{x},\mathbf{y})\sim p_{XY,N}(\mathbf{x},\mathbf{y})}\left[\frac{1}{N}\sum_{i=1}^{N}\log\left(\frac{e^{T_\theta(\mathbf{x}_i,\mathbf{y}_i)}}{\frac{1}{N}\sum_{j=1}^{N}e^{T_\theta(\mathbf{x}_i,\mathbf{y}_j)}}\right)\right], \tag{30}$$

where $N$ is the batch size and $p_{XY,N}$ denotes the joint distribution of $N$ i.i.d. random variables sampled from $p_{XY}$. CPC provides low variance estimates but it is upper bounded by $\log N$, resulting in a biased estimator.

## B.5 NJEE

The neural joint entropy estimator (NJEE) proposed in [24] is based on a classification task. Let $X_m$ be the $m$-th component of $X$, with $m \leq d$ and $N$ the batch size. $X^k$ is the vector containing the first $k$ components of $X$. Let $\hat{H}_N(X_1)$ be the estimated marginal entropy of the first components in $X$ and let $G_{\theta_m}(X_m|X^{m-1})$ be a neural network classifier, where the input is $X^{m-1}$ and the label used is $X_m$. If $\text{CE}(\cdot)$ is the cross-entropy function, then the MI estimator based on NJEE is defined as

$$I_{NJEE}(X;Y) = \hat{H}_N(X_1) + \sum_{m=2}^{d}\text{CE}(G_{\theta_m}(X_m|X^{m-1})) - \sum_{m=1}^{d}\text{CE}(G_{\theta_m}(X_m|Y,X^{m-1})), \tag{31}$$

where the first two terms of the RHS constitutes the NJEE entropy estimator.

# C   Appendix: Proofs of Lemmas and Theorems

## C.1   Proof of Theorem 3.1

**Theorem 3.1.** *Let $(X,Y) \sim p_{XY}(\mathbf{x},\mathbf{y})$ be a pair of multivariate random variables. Let $\sigma(\cdot)$ be a permutation function such that $p_{\sigma(Y)}(\sigma(\mathbf{y})|\mathbf{x}) = p_Y(\mathbf{y})$ and $T : \text{dom}(X) \times \text{dom}(Y) \to \mathbb{R}$. Let $f^*$ be the Fenchel conjugate of $f : \mathbb{R}_+ \to \mathbb{R}$, a convex lower semicontinuous function that satisfies $f(1) = 0$ with derivative $f'$. If $\mathcal{J}_f(T)$ is a value function defined as*

$$\mathcal{J}_f(T) = \mathbb{E}_{(\mathbf{x},\mathbf{y})\sim p_{XY}(\mathbf{x},\mathbf{y})}\left[T(\mathbf{x},\mathbf{y}) - f^*\Big(T(\mathbf{x},\sigma(\mathbf{y}))\Big)\right], \tag{32}$$

*then*

$$\hat{T}(\mathbf{x},\mathbf{y}) = \arg\max_T \mathcal{J}_f(T) = f'\left(\frac{p_{XY}(\mathbf{x},\mathbf{y})}{p_X(\mathbf{x})p_Y(\mathbf{y})}\right), \tag{33}$$

*and*

$$I(X;Y) = I_{fDIME}(X;Y) = \mathbb{E}_{(\mathbf{x},\mathbf{y})\sim p_{XY}(\mathbf{x},\mathbf{y})}\left[\log\Big((f^*)'(\hat{T}(\mathbf{x},\mathbf{y}))\Big)\right]. \tag{34}$$

*Proof.* From the hypothesis, the value function can be rewritten as

$$\mathcal{J}_f(T) = \mathbb{E}_{(\mathbf{x},\mathbf{y})\sim p_{XY}(\mathbf{x},\mathbf{y})}\left[T(\mathbf{x},\mathbf{y})\right] - \mathbb{E}_{(\mathbf{x},\mathbf{y})\sim p_X(\mathbf{x})p_Y(\mathbf{y})}\left[f^*\Big(T(\mathbf{x},\mathbf{y})\Big)\right]. \tag{35}$$

The thesis follows immediately from Lemma 1 of [18]. Indeed, the $f$-divergence $D_f$ can be expressed in terms of its lower bound via Fenchel convex duality

$$D_f(P||Q) \geq \sup_{T\in\mathbb{R}}\left\{\mathbb{E}_{x\sim p(x)}\big[T(x)\big] - \mathbb{E}_{x\sim q(x)}\big[f^*(T(x))\big]\right\}, \tag{36}$$

where $T : \mathcal{X} \to \mathbb{R}$ and $f^*$ is the Fenchel conjugate of $f$ defined as

$$f^*(t) := \sup_{u\in\mathbb{R}}\{ut - f(u)\}. \tag{37}$$

Therein, it was shown that the bound in (36) is tight for optimal values of $T(x)$ and it takes the following form

$$\hat{T}(x) = f'\left(\frac{p(x)}{q(x)}\right), \tag{38}$$

where $f'$ is the derivative of $f$.

The MI $I(X;Y)$ admits the KL divergence representation

$$I(X;Y) = D_{KL}(p_{XY}||p_X p_Y), \tag{39}$$

and since the inverse of the derivative of $f$ is the derivative of the conjugate $f^*$, the density ratio can be rewritten in terms of the optimum discriminator $\hat{T}$

$$(f')^{-1}(\hat{T}(\mathbf{x}, \mathbf{y})) = (f^*)'(\hat{T}(\mathbf{x}, \mathbf{y})) = \frac{p_{XY}(\mathbf{x}, \mathbf{y})}{p_X(\mathbf{x})p_Y(\mathbf{y})}. \tag{40}$$

$f$-DIME finally reads as follows

$$I_{fDIME}(X;Y) = \mathbb{E}_{(\mathbf{x}, \mathbf{y}) \sim p_{XY}(\mathbf{x}, \mathbf{y})} \left[ \log \left( (f^*)'(\hat{T}(\mathbf{x}, \mathbf{y})) \right) \right]. \tag{41}$$

$\square$

## C.2  Proof of Lemma 3.2

**Lemma 3.2.** *Let the discriminator $T(\cdot)$ be with enough capacity, i.e., in the non parametric limit. Consider the problem*

$$\hat{T} = \arg\max_T \mathcal{J}_f(T) \tag{42}$$

*where*

$$\mathcal{J}_f(T) = \mathbb{E}_{(\mathbf{x}, \mathbf{y}) \sim p_{XY}(\mathbf{x}, \mathbf{y})} \left[ T(\mathbf{x}, \mathbf{y}) \right] - \mathbb{E}_{(\mathbf{x}, \mathbf{y}) \sim p_X(\mathbf{x})p_Y(\mathbf{y})} \left[ f^* \left( T(\mathbf{x}, \mathbf{y}) \right) \right], \tag{43}$$

*and the update rule based on the gradient descent method*

$$T^{(n+1)} = T^{(n)} + \mu \nabla \mathcal{J}_f(T^{(n)}). \tag{44}$$

*If the gradient descent method converges to the global optimum $\hat{T}$, the mutual information estimator*

$$I(X;Y) = I_{fDIME}(X;Y) = \mathbb{E}_{(\mathbf{x}, \mathbf{y}) \sim p_{XY}(\mathbf{x}, \mathbf{y})} \left[ \log \left( (f^*)'(\hat{T}(\mathbf{x}, \mathbf{y})) \right) \right]. \tag{45}$$

*converges to the real value of the mutual information $I(X;Y)$.*

*Proof.* For convenience of notation, let the instantaneous MI be the random variable defined as

$$i(X;Y) := \log \left( \frac{p_{XY}(\mathbf{x}, \mathbf{y})}{p_X(\mathbf{x})p_Y(\mathbf{y})} \right). \tag{46}$$

It is straightforward to notice that the MI corresponds to the expected value of $i(X;Y)$ over the joint distribution $p_{XY}$. The solution to (42) is given by (6) of Theorem 3.1. Let $\delta^{(n)} = \hat{T} - T^{(n)}$ be the displacement between the optimum discriminator $\hat{T}$ and the obtained one $T^{(n)}$ at the iteration $n$, then

$$\hat{i}_{n,fDIME}(X;Y) = \log \left( (f^*)'(T^{(n)}(\mathbf{x}, \mathbf{y})) \right) = \log \left( R^{(n)}(\mathbf{x}, \mathbf{y}) \right), \tag{47}$$

where $R^{(n)}(\mathbf{x}, \mathbf{y})$ represents the estimated density ratio at the $n$-th iteration and it is related with the optimum ratio $\hat{R}(\mathbf{x}, \mathbf{y})$ as follows

$$\begin{aligned} \hat{R} - R^{(n)} &= (f^*)'(\hat{T}) - (f^*)'(T^{(n)}) \\ &= (f^*)'(\hat{T}) - (f^*)'(\hat{T} - \delta^{(n)}) \\ &\simeq \delta^{(n)} \cdot \left[ (f^*)''(\hat{T} - \delta^{(n)}) \right], \end{aligned} \tag{48}$$

where the last step follows from a first order Taylor expansion in $\hat{T} - \delta^{(n)}$. Therefore,

$$
\begin{aligned}
\hat{i}_{n,fDIME}(X;Y) &= \log\big(R^{(n)}\big) \\
&= \log\Bigg( (\hat{R})\bigg( 1 - \delta^{(n)} \cdot \frac{(f^*)''(\hat{T} - \delta^{(n)})}{(f^*)'(\hat{T})} \bigg) \Bigg) \\
&= i(X;Y) + \log\Bigg( 1 - \delta^{(n)} \cdot \frac{(f^*)''(\hat{T} - \delta^{(n)})}{(f^*)'(\hat{T})} \Bigg).
\end{aligned}
\tag{49}
$$

If the gradient descent method converges towards the optimum solution $\hat{T}$, $\delta^{(n)} \to 0$ and

$$
\begin{aligned}
\hat{i}_{n,fDIME}(X;Y) &\simeq i(X;Y) - \delta^{(n)} \cdot \left[ \frac{(f^*)''(\hat{T} - \delta^{(n)})}{(f^*)'(\hat{T})} \right] \\
&\simeq i(X;Y) - \delta^{(n)} \cdot \left[ \frac{(f^*)''(\hat{T})}{(f^*)'(\hat{T})} \right] \\
&= i(X;Y) - \delta^{(n)} \cdot \left[ \frac{\mathrm{d}}{\mathrm{d}T} \log\big((f^*)'(T)\big) \bigg|_{T=\hat{T}} \right],
\end{aligned}
\tag{50}
$$

where the RHS is itself a first order Taylor expansion of the instantaneous MI in $\hat{T}$. In the asymptotic limit ($n \to +\infty$), it holds also for the expected values that

$$
|I(X;Y) - \hat{I}_{n,fDIME}(X;Y)| \to 0.
\tag{51}
$$

$\square$

### C.3 Proof of Lemma 4.1

**Lemma 4.1.** *Let $\hat{R} = p_{XY}(\mathbf{x}, \mathbf{y}) / (p_X(\mathbf{x}) p_Y(\mathbf{y}))$ and assume $Var_{p_{XY}}[\log \hat{R}]$ exists. Let $p_{XY}^M$ be the empirical distribution of $M$ i.i.d. samples from $p_{XY}$ and let $\mathbb{E}_{p_{XY}^M}$ denote the sample average over $p_{XY}^M$. Then, under the randomness of the sampling procedure, it holds that*

$$
Var_{p_{XY}}\big[\mathbb{E}_{p_{XY}^M}[\log \hat{R}]\big] \leq \frac{4H^2(p_{XY}, p_X p_Y)\|\hat{R}\|_\infty - I^2(X;Y)}{M}
\tag{52}
$$

*where $H^2$ is the Hellinger distance squared defined as*

$$
H^2(p, q) = \int_{\mathbf{x}} \left( \sqrt{p(\mathbf{x})} - \sqrt{q(\mathbf{x})} \right)^2 \mathrm{d}\mathbf{x},
\tag{53}
$$

*and the infinity norm is defined as $\|f(x)\|_\infty := \sup_{x \in \mathbb{R}} |f(x)|$.*

*Proof.* Consider the variance of $\hat{R}(\mathbf{x}, \mathbf{y})$ when $(\mathbf{x}, \mathbf{y}) \sim p_{XY}(\mathbf{x}, \mathbf{y})$, then

$$
\mathrm{Var}_{p_{XY}}[\log \hat{R}] = \mathbb{E}_{p_{XY}}\left[ \left( \log \frac{p_{XY}}{p_X p_Y} \right)^2 \right] - \left( \mathbb{E}_{p_{XY}}\left[ \log \frac{p_{XY}}{p_X p_Y} \right] \right)^2.
\tag{54}
$$

The power of the log-density ratio is upper bounded as follows (see the approach of Lemma 8.3 in [29])

$$
\mathbb{E}_{p_{XY}}\left[ \left( \log \frac{p_{XY}}{p_X p_Y} \right)^2 \right] \leq 4H^2(p_{XY}, p_X p_Y) \left\| \frac{p_{XY}}{p_X p_Y} \right\|_\infty,
\tag{55}
$$

while the mean squared is the ground-truth MI squared, thus

$$
\mathrm{Var}_{p_{XY}}[\log \hat{R}] \leq 4H^2(p_{XY}, p_X p_Y) \left\| \frac{p_{XY}}{p_X p_Y} \right\|_\infty - I^2(X;Y).
\tag{56}
$$

Finally, the variance of the mean of $M$ i.i.d. random variables yields the thesis

$$
\mathrm{Var}_{p_{XY}}\big[\mathbb{E}_{p_{XY}^M}[\log \hat{R}]\big] = \frac{\mathrm{Var}_{p_{XY}}[\log \hat{R}]}{M} \leq \frac{4H^2(p_{XY}, p_X p_Y) \left\| \frac{p_{XY}}{p_X p_Y} \right\|_\infty - I^2(X;Y)}{M}.
\tag{57}
$$

$\square$

## C.4 Proof of Lemma 4.2

**Lemma 4.2.** *Let $\hat{R}$ be the optimal density ratio and let $X \sim \mathcal{N}(0, \sigma_X^2)$ and $N \sim \mathcal{N}(0, \sigma_N^2)$ be uncorrelated scalar Gaussian random variables such that $Y = X + N$. Assume $Var_{p_{XY}}[\log \hat{R}]$ exists. Let $p_{XY}^M$ be the empirical distribution of $M$ i.i.d. samples from $p_{XY}$ and let $\mathbb{E}_{p_{XY}^M}$ denote the sample average over $p_{XY}^M$. Then, under the randomness of the sampling procedure, it holds that*

$$Var_{p_{XY}}\left[\mathbb{E}_{p_{XY}^M}[\log \hat{R}]\right] = \frac{1 - e^{-2I(X;Y)}}{M}. \tag{58}$$

*Proof.* From the hypothesis, the density ratio can be rewritten as $\hat{R} = p_N(y - x)/p_Y(y)$ and the output variance is clearly equal to $\sigma_Y^2 = \sigma_X^2 + \sigma_N^2$. Notice that this is equivalent of having correlated random variables $X$ and $Y$ with correlation coefficient $\rho$, since it is enough to study the case $\sigma_X = \rho$ and $\sigma_N = \sqrt{1 - \rho^2}$.

It is easy to verify via simple calculations that

$$
\begin{aligned}
I(X;Y) &= \mathbb{E}_{p_{XY}}[\log \hat{R}] \\
&= \log \frac{\sigma_Y}{\sigma_N} + \mathbb{E}_{p_{XY}}\left[\frac{y^2}{2\sigma_Y^2} - \frac{(y - x)^2}{2\sigma_N^2}\right] \\
&= \cdots = \log \frac{\sigma_Y}{\sigma_N} = \frac{1}{2}\log\left(1 + \frac{\sigma_X^2}{\sigma_N^2}\right) = -\frac{1}{2}\log(1 - \rho^2).
\end{aligned} \tag{59}
$$

Similarly,

$$
\begin{aligned}
Var_{p_{XY}}\left[\log \hat{R}\right] &= \mathbb{E}_{p_{XY}}\left[\left(\log\left(\frac{\sigma_Y}{\sigma_N}\right) + \frac{y^2}{2\sigma_Y^2} - \frac{(y - x)^2}{2\sigma_N^2}\right)^2\right] - I^2(X;Y) \\
&= \frac{1}{4}\mathbb{E}_{p_{XY}}\left[\left(\frac{y - x}{\sigma_N}\right)^4 + \left(\frac{y}{\sigma_Y}\right)^4 - 2\left(\frac{y}{\sigma_Y}\right)^2\left(\frac{y - x}{\sigma_N}\right)^2\right] \\
&= \cdots = \text{Kurt}[Z]\left(\frac{1}{2} - \frac{\sigma_N^2}{2\sigma_Y^2}\right) - \frac{\sigma_X^2}{2\sigma_Y^2} \\
&= \frac{\sigma_X^2}{\sigma_Y^2} = 1 - \frac{\sigma_N^2}{\sigma_Y^2} = 1 - e^{-2I(X;Y)} = \rho^2,
\end{aligned} \tag{60}
$$

where the last steps use the fact that the Kurtosis of a normal distribution is 3 and that the MI can be expressed as in (59). Finally, the variance of the mean of $M$ i.i.d. random variables yields the thesis

$$Var_{p_{XY}}\left[\mathbb{E}_{p_{XY}^M}[\log \hat{R}]\right] = \frac{Var_{p_{XY}}[\log \hat{R}]}{M}. \tag{61}$$

If $X$ and $N$ are multivariate Gaussians with diagonal covariance matrices $\rho^2 \mathbb{I}_{d \times d}$ and $(1 - \rho^2)\mathbb{I}_{d \times d}$, the results for both the MI and variance in the scalar case are simply multiplied by $d$. ☐

## C.5 Proof of Lemma 5.1

**Lemma 5.1.** *Let $(\mathbf{x}_i, \mathbf{y}_i)$, $\forall i \in \{1, \ldots, N\}$, be $N$ data points. Let $\mathcal{J}_f(T)$ be the value function in (5). Let $\mathcal{J}_f^\pi(T)$ and $\mathcal{J}_f^\sigma(T)$ be numerical implementations of $\mathcal{J}_f(T)$ using a random permutation and a random derangement of $\mathbf{y}$, respectively. Denote with $K$ the number of points $\mathbf{y}_k$, with $k \in \{1, \ldots, N\}$, in the same position after the permutation (i.e., the fixed points). Then*

$$\mathcal{J}_f^\pi(T) \leq \frac{N - K}{N}\mathcal{J}_f^\sigma(T). \tag{62}$$

*Proof.* Define $\mathcal{J}_f^\pi(T)$ as the Monte Carlo implementation of $\mathcal{J}_f(T)$ when using the permutation function $\pi(\cdot)$

$$\mathcal{J}_f^\pi(T) = \frac{1}{N}\sum_{i=1}^{N} T(\mathbf{x}_i, \mathbf{y}_i) - \frac{1}{N}\sum_{i=1}^{N} f^*\big(T(\mathbf{x}_i, \mathbf{y}_j)\big), \tag{63}$$

where the pair $(\mathbf{x}_i, \mathbf{y}_j)$ is obtained via a random permutation of the elements of $\mathbf{y}$ as $j = \pi(i)$, $\forall i \in \{1, \ldots, N\}$. Since $K$ is a non-negative integer representing the number of fixed points $i = \pi(i)$, the value function can be rewritten as

$$\mathcal{J}_f^\pi(T) = \frac{1}{N} \sum_{i=1}^{N} T(\mathbf{x}_i, \mathbf{y}_i) - \frac{1}{N} \sum_{i=1}^{K} f^* \big( T(\mathbf{x}_i, \mathbf{y}_i) \big) - \frac{1}{N} \sum_{i=1}^{N-K} f^* \big( T(\mathbf{x}_i, \mathbf{y}_{j \neq i}) \big), \quad (64)$$

which can also be expressed as

$$\mathcal{J}_f^\pi(T) = \frac{1}{N} \sum_{i=1}^{K} T(\mathbf{x}_i, \mathbf{y}_i) + \frac{1}{N} \sum_{i=1}^{N-K} T(\mathbf{x}_i, \mathbf{y}_i) - \frac{1}{N} \sum_{i=1}^{K} f^* \big( T(\mathbf{x}_i, \mathbf{y}_i) \big) - \frac{1}{N} \sum_{i=1}^{N-K} f^* \big( T(\mathbf{x}_i, \mathbf{y}_{j \neq i}) \big). \quad (65)$$

In (65) it is possible to recognize that the second and last term of the RHS constitutes the numerical implementation of $\mathcal{J}_f(T)$ using a derangement strategy on $N - K$ elements, so that

$$\mathcal{J}_f^\pi(T) = \frac{1}{N} \sum_{i=1}^{K} T(\mathbf{x}_i, \mathbf{y}_i) - \frac{1}{N} \sum_{i=1}^{K} f^* \big( T(\mathbf{x}_i, \mathbf{y}_i) \big) + \frac{N-K}{N} \mathcal{J}_f^\sigma(T). \quad (66)$$

However, by definition of Fenchel conjugate

$$\frac{1}{N} \sum_{i=1}^{K} T(\mathbf{x}_i, \mathbf{y}_i) - f^* \big( T(\mathbf{x}_i, \mathbf{y}_i) \big) \leq 0, \quad (67)$$

since for $t = 1$

$$u - f^*(u) \leq u - (ut - f(t)) = f(1) = 0. \quad (68)$$

Hence, we can conclude that

$$\mathcal{J}_f^\pi(T) \leq \frac{N-K}{N} \mathcal{J}_f^\sigma(T). \quad (69)$$

$\square$

## C.6 Proof of Theorem 5.2

**Theorem 5.2.** *Let the discriminator $D(\cdot)$ be with enough capacity. Let $N$ be the batch size and $f$ be the generator of the KL divergence. Let $\mathcal{J}_{KL}^\pi(D)$ be defined as*

$$\mathcal{J}_{KL}^\pi(D) = \mathbb{E}_{(\mathbf{x},\mathbf{y}) \sim p_{XY}(\mathbf{x},\mathbf{y})} \left[ \log \Big( D(\mathbf{x}, \mathbf{y}) \Big) - f^* \Big( \log \Big( D(\mathbf{x}, \pi(\mathbf{y})) \Big) \Big) \right]. \quad (70)$$

*Denote with $K$ the number of indices in the same position after the permutation (i.e., the fixed points), and with $R(\mathbf{x}, \mathbf{y})$ the density ratio in (2). Then,*

$$\hat{D}(\mathbf{x}, \mathbf{y}) = \arg\max_D \mathcal{J}_{KL}^\pi(D) = \frac{N R(\mathbf{x}, \mathbf{y})}{K R(\mathbf{x}, \mathbf{y}) + N - K}. \quad (71)$$

*Proof.* The idea of the proof is to express $\mathcal{J}_{KL}^\pi(D)$ via Monte Carlo approximation, in order to rearrange fixed points, and then go back to Lebesgue integration. The value function $\mathcal{J}_{KL}(D)$ can be written as

$$\mathcal{J}_{KL}(D) = \mathbb{E}_{(\mathbf{x},\mathbf{y}) \sim p_{XY}(\mathbf{x},\mathbf{y})} \left[ \log \big( D(\mathbf{x}, \mathbf{y}) \big) \right] - \mathbb{E}_{(\mathbf{x},\mathbf{y}) \sim p_X(\mathbf{x}) p_Y(\mathbf{y})} \left[ D(\mathbf{x}, \mathbf{y}) \right] + 1. \quad (72)$$

Similarly to (64), we can express $\mathcal{J}_{KL}^\pi(D)$ as

$$\mathcal{J}_{KL}^\pi(D) = \frac{1}{N} \sum_{i=1}^{N} \log \big( D(\mathbf{x}_i, \mathbf{y}_i) \big) - \frac{1}{N} \sum_{i=1}^{K} D(\mathbf{x}_i, \mathbf{y}_i) - \frac{1}{N} \sum_{i=1}^{N-K} D(\mathbf{x}_i, \mathbf{y}_{j \neq i}) + 1, \quad (73)$$

where $K$ is the number of fixed points of the permutation $j = \pi(i), \forall i \in \{1, \ldots, N\}$. However, when $N \to \infty$, we can use Lebesgue integration and rewrite (73) as

$$\mathcal{J}_{KL}^\pi(D) = \int_{\mathbf{x}} \int_{\mathbf{y}} \left( p_{XY}(\mathbf{x}, \mathbf{y}) \log \big( D(\mathbf{x}, \mathbf{y}) \big) - \frac{K}{N} p_{XY}(\mathbf{x}, \mathbf{y}) D(\mathbf{x}, \mathbf{y}) \right) \mathrm{d}\mathbf{x} \, \mathrm{d}\mathbf{y}$$
$$- \int_{\mathbf{x}} \int_{\mathbf{y}} \frac{N-K}{N} p_X(\mathbf{x}) p_Y(\mathbf{y}) D(\mathbf{x}, \mathbf{y}) \, \mathrm{d}\mathbf{x} \, \mathrm{d}\mathbf{y} + 1. \quad (74)$$

To maximize $\mathcal{J}_{KL}^\pi(D)$, it is enough to take the derivative of the integrand with respect to $D$ and equate it to 0, yielding the following equation in $D$

$$\frac{p_{XY}(\mathbf{x}, \mathbf{y})}{D(\mathbf{x}, \mathbf{y})} - \frac{K}{N} p_{XY}(\mathbf{x}, \mathbf{y}) - \frac{N - K}{N} p_X(\mathbf{x}) p_Y(\mathbf{y}) = 0. \tag{75}$$

Solving for $D$ leads to the thesis

$$\hat{D}(\mathbf{x}, \mathbf{y}) = \frac{NR(\mathbf{x}, \mathbf{y})}{KR(\mathbf{x}, \mathbf{y}) + N - K}, \tag{76}$$

since $\mathcal{J}_{KL}^\pi(\hat{D})$ is a maximum being the second derivative w.r.t. $D$ a non-positive function. □

## C.7 Proof of Corollary 5.3

**Corollary 5.3** (Permutation bound). *Let KL-DIME be the estimator obtained via iterative optimization of $\mathcal{J}_{KL}^\pi(D)$, using a batch of size $N$ every training step. Then,*

$$I_{KL-DIME}^\pi := \mathbb{E}_{(\mathbf{x}, \mathbf{y}) \sim p_{XY}(\mathbf{x}, \mathbf{y})} \left[ \log \left( \hat{D}(\mathbf{x}, \mathbf{y}) \right) \right] < \log(N). \tag{77}$$

*Proof.* Theorem 5.2 implies that, when the batch size is much larger than the density ratio ($N >> R$), then the discriminator's output converges to the density ratio. Indeed,

$$\lim_{N \to \infty} \hat{D}(\mathbf{x}, \mathbf{y}) = \lim_{N \to \infty} \frac{NR(\mathbf{x}, \mathbf{y})}{KR(\mathbf{x}, \mathbf{y}) + N - K} = R(\mathbf{x}, \mathbf{y}). \tag{78}$$

Instead, when the density ratio is much larger than the batch size ($R >> N$), then the discriminator's output converges to a constant, in particular

$$\lim_{R \to \infty} \hat{D}(\mathbf{x}, \mathbf{y}) = \lim_{R \to \infty} \frac{NR(\mathbf{x}, \mathbf{y})}{KR(\mathbf{x}, \mathbf{y}) + N - K} = \frac{N}{K}. \tag{79}$$

However, from Lemma 5.4, it is true that $K = 1$ on average. Therefore, an iterative optimization algorithm leads to an upper-bounded discriminator, since

$$\hat{D}(\mathbf{x}, \mathbf{y}) < N, \tag{80}$$

which implies the thesis. □

## C.8 Proof of Lemma 5.4

**Lemma 5.4** (see [30]). *The average number of fixed points in a random permutation $\pi(\cdot)$ is equal to 1.*

*Proof.* Let $\pi(\cdot)$ be a random permutation on $\{1, \ldots, N\}$. Let the random variable $X$ represent the number of fixed points (i.e., the number of cycles of length 1) of $\pi(\cdot)$. We define $X = X_1 + X_2 + \cdots + X_N$, where $X_i = 1$ when $\pi(i) = i$, and 0 otherwise. $\mathbb{E}[X]$ is computed by exploiting the linearity property of expectation. Trivially,

$$\mathbb{E}[X_i] = \mathbb{P}[\pi(i) = i] = \frac{1}{N}, \tag{81}$$

which implies

$$\mathbb{E}[X] = \sum_{i=1}^{N} \frac{1}{N} = 1. \tag{82}$$

□

Table 1: Neural architectures comparison.

| | Joint | Separable | Deranged |
|---|---|---|---|
| Input | $N$ pairs $(x, y) \sim p_{XY}$ 
 $N(N-1)$ pairs $(x, \tilde{y}) \sim p_X p_Y$ | $N$ pairs $(x, y) \sim p_{XY}$ 
 $N$ pairs $(x, \tilde{y}) \sim p_X p_Y$ | $N$ pairs $(x, y) \sim p_{XY}$ 
 $N$ pairs $(x, \tilde{y}) \sim p_X p_Y$ |
| Nr. NNs | 1 | 2 | 1 |
| Complexity | $\Omega(N^2)$ | $\Omega(N)$ | $\Omega(N)$ |

# D   Appendix: Experimental Details

## D.1   Multivariate Linear and Nonlinear Gaussians Experiments

In this section, we show supplementary results for the linear and cubic Gaussian experiments. The implemented neural network architectures are: *joint*, *separable*, *deranged*, and the architecture of NJEE, referred to as *ad hoc*. See Tab. 1 for a schematic about the architectures.

**Joint architecture**. The *joint* architecture is a feed-forward fully connected neural network with an input size equal to twice the dimension of the samples distribution ($2d$), one output neuron, and two hidden layers of 256 neurons each. The activation function utilized in each layer (except from the last one) is ReLU. The number of realizations $(\mathbf{x}, \mathbf{y})$ fed as input of the neural network for each training iteration is $N^2$, obtained as all the combinations of the samples $\mathbf{x}$ and $\mathbf{y}$ drawn from $p_{XY}(\mathbf{x}, \mathbf{y})$.

**Separable architecture**. The *separable* architecture comprises two feed-forward neural networks, each one with an input size equal to $d$, output layer containing 32 neurons and 2 hidden layers with 256 neurons each. The ReLU activation function is used in each layer (except from the last one). The first network is fed in with $N$ realizations of $X$, while the second one with $N$ realizations of $Y$.

**Deranged architecture**. The *deranged* architecture is a feed-forward fully connected neural network with an input size equal to twice the dimension of the samples distribution ($2d$), one output neuron, and two hidden layers of 256 neurons each. The activation function utilized in each layer (except from the last one) is ReLU. The number of realizations $(\mathbf{x}, \mathbf{y})$ the neural network is fed with is $2N$ for each training iteration: $N$ realizations drawn from $p_{XY}(\mathbf{x}, \mathbf{y})$ and $N$ realizations drawn from $p_X(\mathbf{x})p_Y(\mathbf{y})$ using the derangement procedure described in Sec. 5.
The architecture *deranged* is not used for $I_{CPC}$ because in (30) the summation at the denominator of the argument of the logarithm would require neural network predictions corresponding to the inputs $(\mathbf{x}_i, \mathbf{y}_j)$, $\forall i, j \in \{1, \ldots, N\}$ with $i \neq j$.

**Ad hoc architecture**. The *NJEE* MI estimator comprises $2d - 1$ feed-forward neural networks. Each neural network is composed by an input layer with size between 1 and $2d - 1$, an output layer containing $N - k$ neurons, with $k \in \mathbb{N}$ small, and 2 hidden layers with 256 neurons each. The ReLU activation function is used in each layer (except from the last one). We implemented a Pytorch [31] version of the code produced by the authors of [24] [3], to unify NJEE with all the other MI estimators.

Each neural estimator is trained using Adam optimizer [32], with learning rate $5 \times 10^{-4}$, $\beta_1 = 0.9$, $\beta_2 = 0.999$. The batch size is initially set to $N = 64$.
For the *Gaussian* setting, we sample a 20-dimensional Gaussian distribution to obtain $\mathbf{x}$ and $\mathbf{n}$ samples, independently. Then, we compute $\mathbf{y}$ as linear combination of $\mathbf{x}$ and $\mathbf{n}$: $\mathbf{y} = \rho \mathbf{x} + \sqrt{1 - \rho^2} \mathbf{n}$, where $\rho$ is the correlation coefficient. For the *cubic* setting, the nonlinear transformation $\mathbf{y} \mapsto \mathbf{y}^3$ is applied to the Gaussian samples. During the training procedure, every $4k$ iterations, the target value of the MI is increased by $2\, nats$, for 5 times, obtaining a target staircase with 5 steps. The change in target MI is obtained by increasing $\rho$, that affects the true MI according to

$$I(X; Y) = -\frac{d}{2} \log(1 - \rho^2). \tag{83}$$

---

[3]https://github.com/YuvalShalev/NJEE

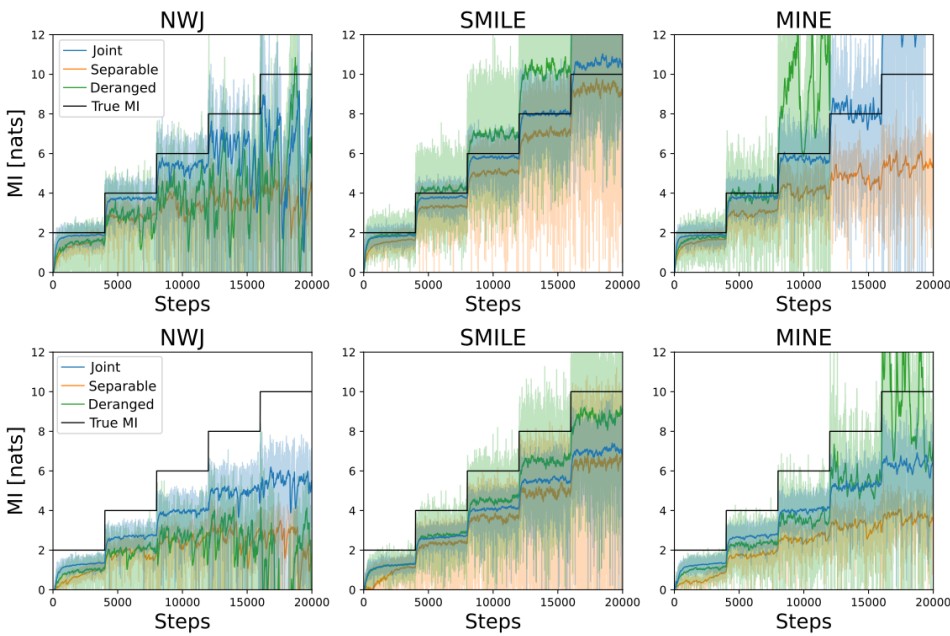

Figure 6: NWJ, SMILE ($\tau = \infty$), and MINE MI estimation comparison with $d = 20$ and $N = 64$. The *Gaussian* setting is represented in the top row, while the *cubic* setting is shown in the bottom row.

#### D.1.1  Supplementary Analysis of the MI Estimators Performance

Additional plots reporting the MI estimates obtained from MINE, NWJ, and SMILE with $\tau = \infty$, are outlined in Fig. 6. The variance attained by these algorithms exponentially increases as the true MI grows, as stated in (10).

We report in Fig. 7 the behavior we obtained for $I_{SMILE}$ when the training of the neural network is performed by using the cost function in (29). The training diverges during the first steps when $\tau = 1$ and $\tau = 5$. Differently, when $\tau = \infty$, $I_{SMILE}$ corresponds to $I_{MINE}$ (without the moving average improvement), therefore the MI estimate does not diverge. Interestingly, by comparing $I_{SMILE}$ ($\tau = \infty$) trained with the JS divergence and with the MINE cost function (in Fig. 6 and Fig. 7, respectively), the variance of the latter case is significantly higher. Hence, the JS maximization trick seems to have an impact in lowering the estimator variance.

#### D.1.2  Analysis for Different Values of $d$ and $N$

The class of $f$-DIME estimators is robust to changes in $d$ and $N$, as the estimators' variance decreases (see (58) and Fig. 15) when $N$ increases and their achieved bias is not significantly influenced by the choice of $d$. Differently, $I_{NJEE}$ and $I_{CPC}$ are highly affected by variations of those parameters, as described in Fig. 2 and Fig. 10. More precisely, $I_{CPC}$ is not strongly influenced by a change of $d$, but the bias significantly increases as the batch size diminishes, since the upper bound lowers. $I_{NJEE}$ achieves a higher bias both when $d$ decreases and when $N$ increases w.r.t. the default values $d = 20, N = 64$. In addition, when $d$ is large, the training of $I_{NJEE}$ is not feasible, as it requires a lot of time (see Fig. 5) and memory (as a consequence of the large number of neural networks utilized) requirements. In addition, Fig. 8 illustrates that the time complexity of the joint architecture is $\Omega(N^2)$, while the complexity of the deranged architecture is $\Omega(N)$.

We show the achieved bias, variance, and mean squared error (MSE) corresponding to the three settings reported in Fig. 2, 10, and 11 in Fig. 12, 13, and 14, respectively. The achieved variance is bounded when the estimator used is $I_{KL-DIME}$ or $I_{CPC}$. In particular, Figs. 12, 13, 14, and 15 demonstrate that $I_{KL-DIME}$ satisfies Lemma 4.2.

Additionally, we report the achieved bias, variance and MSE when $d = 20$ and $N$ varies according to Tab. 3. We use the notation $N = [512, 1024]$ to indicate that each cell of the table reports the values corresponding to $N = 512$ and $N = 1024$, with this specific order, inside the brackets. Similarly, we

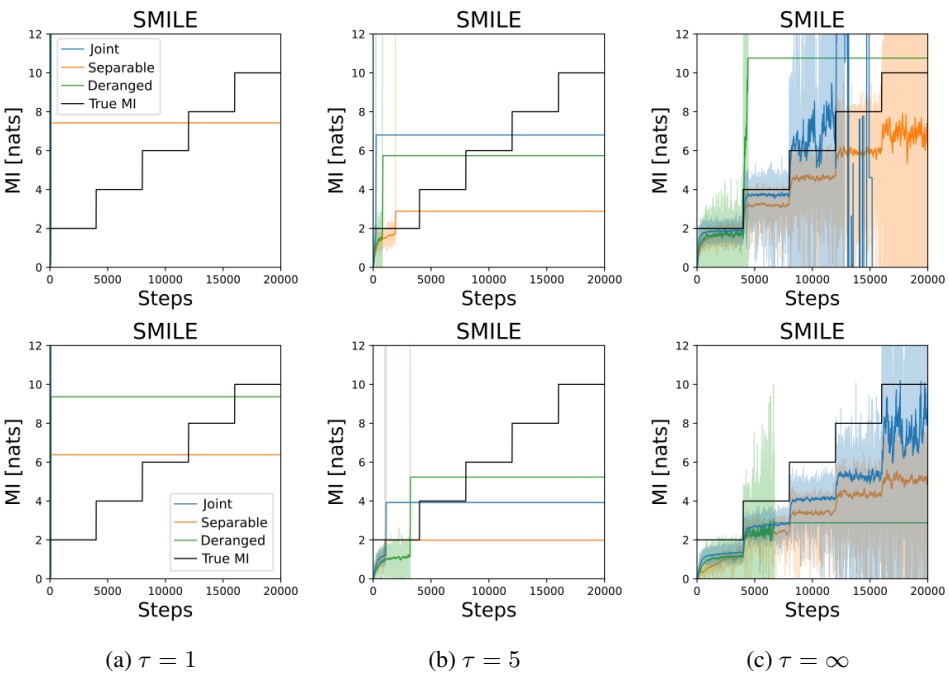

Figure 7: $I_{SMILE}$ behavior for different values of $\tau$, when the JS divergence is not used to train the neural network. The *Gaussian* case is reported in the top row, while the *cubic* case is reported in the bottom row.

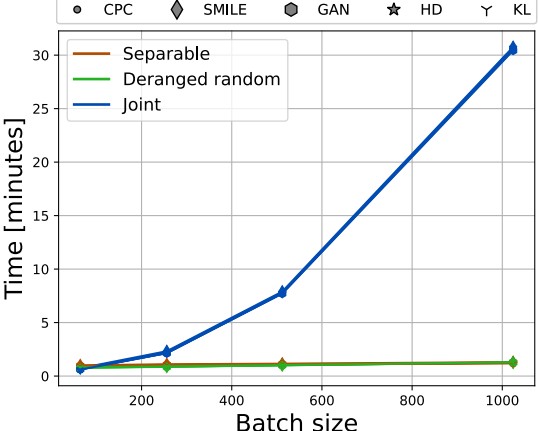

Figure 8: Time requirements comparison to complete the 5-step staircase MI over the batch size. Linear scale.

Table 2: Variance comparison between the VLB MI estimators and $f$-DIME, using the joint architecture, when $d = 5$ and $N = 64$, for the Gaussian setting.

| MI | 2 | 4 | 6 | 8 | 10 |
|---|---|---|---|---|---|
| NWJ | 0.05 | 0.13 | 0.69 | 5.34 | 9.48 |
| MINE | 0.05 | 0.11 | 0.39 | 1.73 | 17.10 |
| SMILE ($\tau = \infty$) | 0.05 | 0.11 | 0.32 | 1.40 | 8.89 |
| GAN-DIME | 0.05 | 0.08 | 0.13 | 0.24 | 0.69 |
| HD-DIME | 0.05 | 0.08 | 0.12 | 0.20 | 0.57 |
| KL-DIME | 0.04 | 0.06 | 0.06 | 0.06 | 0.06 |

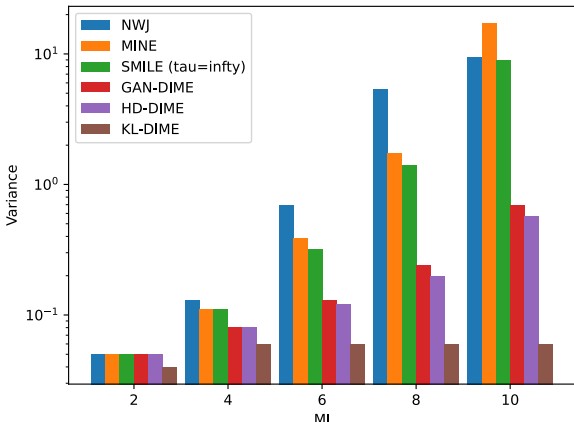

Figure 9: Variance bar plots between the VLB MI estimators and $f$-DIME, using the joint architecture, when $d = 5$ and $N = 64$, for the Gaussian setting.

show the attained bias, variance, and MSE for $d = [5, 10]$ and $N = 64$ in Tab. 5. The achieved bias, variance and MSE shown in Tab. 3 and Tab. 5 motivate that the class of $f$-DIME estimators attains the best values for bias and MSE. Similarly, $I_{KL-DIME}$ obtains the lowest variance, when excluding $I_{CPC}$ from the estimators comparison ($I_{CPC}$ should not be desirable as it is upper bounded). The illustrated results are obtained with the *joint* architecture (except for NJEE) because, when the batch size is small, such an architecture achieves slightly better results than the *deranged* one, as it approximates the expectation over the product of marginals with more samples.

The variance of the $f$-DIME estimators achieved in the Gaussian setting when $N$ ranges from $64$ to $1024$ is reported in Fig. 15. The behavior shown in such a figure demonstrates what is stated in Lemma 4.1, i.e., the variance of the $f$-DIME estimators varies as $\frac{1}{N}$.

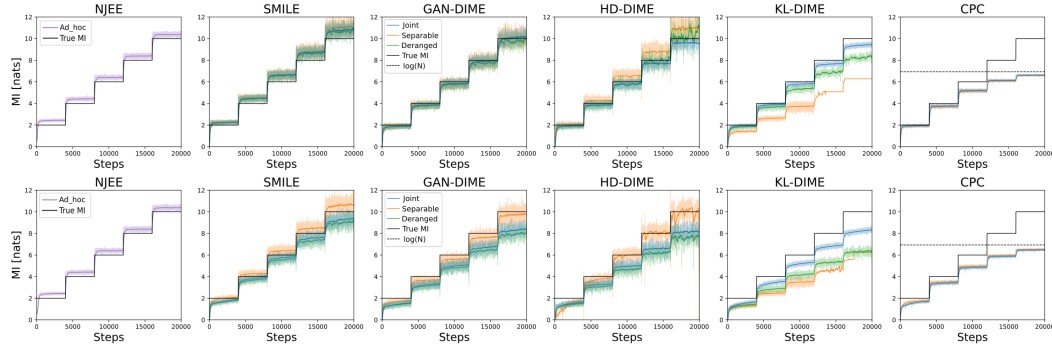

Figure 10: Staircase MI estimation comparison for $d = 20$ and $N = 1024$. The *Gaussian* case is reported in the top row, while the *cubic* case is shown in the bottom row.

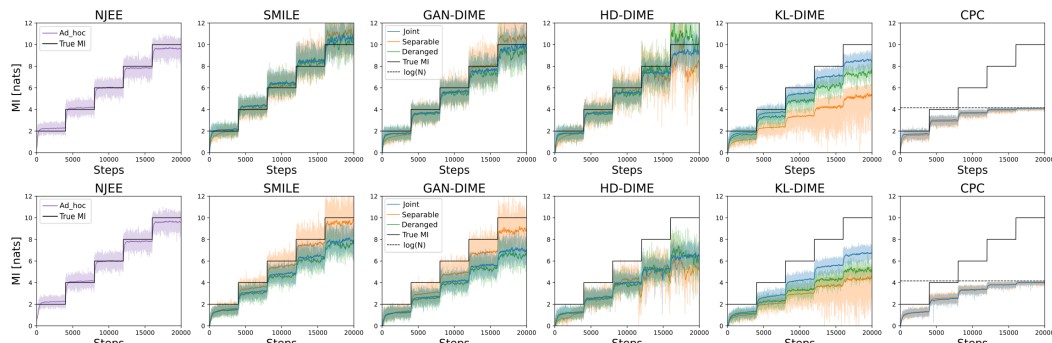

Figure 11: Staircase MI estimation comparison for $d = 20$ and $N = 64$. The *Gaussian* case is reported in the top row, while the *cubic* case is shown in the bottom row.

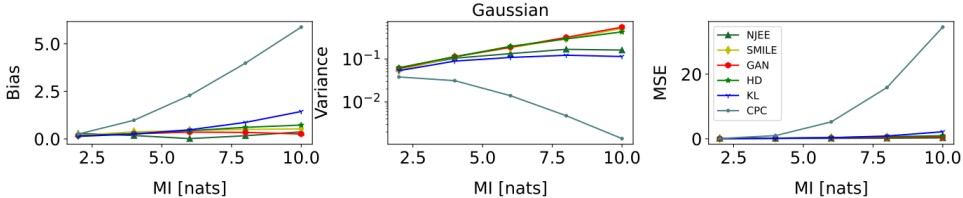

Figure 12: Bias, variance, and MSE comparison between estimators, using the joint architecture for the *Gaussian* case with $d = 20$ and $N = 64$.

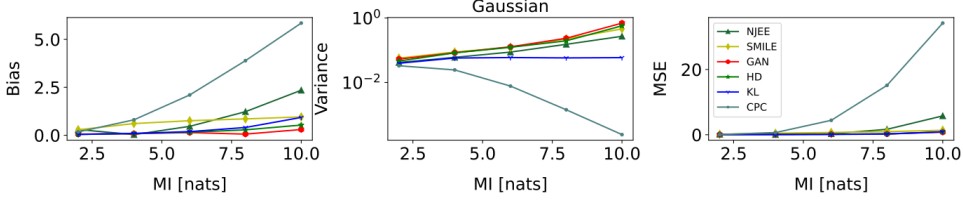

Figure 13: Bias, variance, and MSE comparison between estimators, using the joint architecture for the *Gaussian* case with $d = 5$ and $N = 64$.

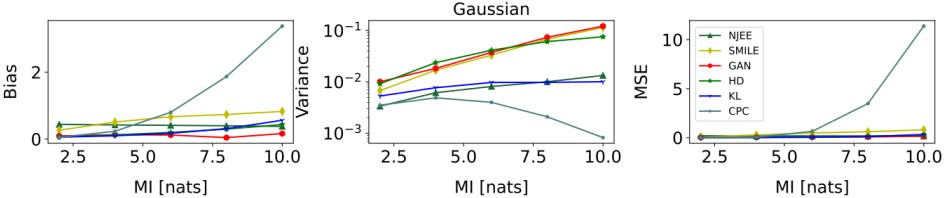

Figure 14: Bias, variance, and MSE comparison between estimators, using the joint architecture for the *Gaussian* case with $d = 20$ and $N = 1024$.

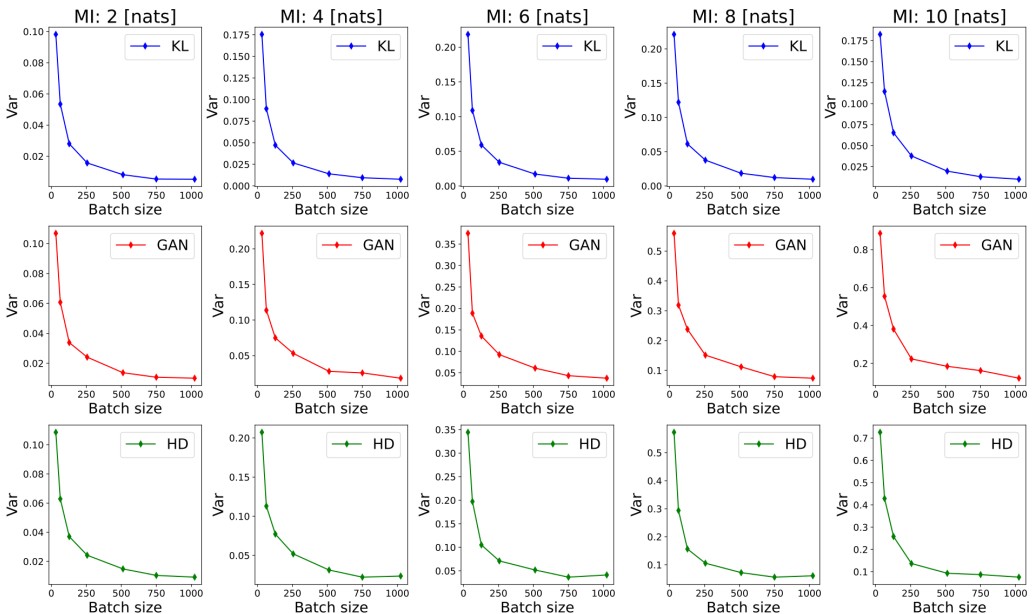

Figure 15: Variance of the $f$-DIME estimators corresponding to different values of batch size.

Table 3: Bias (B), variance (V), and MSE (M) of the MI estimators using the joint architecture, when $d = 20$ and $N = [512, 1024]$, for the Gaussian setting. Each $f$-DIME estimator is abbreviated to $f$-D.

| | | Gaussian | | | | |
|---|---|---|---|---|---|---|
| | MI | 2 | 4 | 6 | 8 | 10 |
| B | NJEE | [0.42, 0.44] | [0.40, 0.42] | [0.37, 0.41] | [0.34, 0.40] | [0.32, 0.38] |
| | SMILE | [0.25, 0.27] | [0.48, 0.51] | [0.64, 0.67] | [0.74, 0.73] | [0.86, 0.83] |
| | GAN-D | [0.11, 0.09] | [0.15, 0.13] | [**0.16**, **0.12**] | [**0.14**, **0.04**] | [**0.01**, **0.16**] |
| | HD-D | [0.08, 0.07] | [0.15, 0.12] | [0.24, 0.20] | [0.37, 0.30] | [0.47, 0.43] |
| | KL-D | [**0.07**, 0.06] | [**0.12**, **0.10**] | [0.21, 0.17] | [0.38, 0.31] | [0.69, 0.56] |
| | CPC | [0.08, **0.05**] | [0.34, 0.23] | [1.07, 0.80] | [2.32, 1.87] | [3.96, 3.37] |
| V | NJEE | [0.01, 0.00] | [0.01, 0.01] | [0.02, 0.01] | [0.02, 0.01] | [0.02, 0.01] |
| | SMILE | [0.01, 0.01] | [0.03, 0.02] | [0.06, 0.03] | [0.11, 0.07] | [0.17, 0.11] |
| | GAN-D | [0.01, 0.01] | [0.03, 0.02] | [0.06, 0.04] | [0.11, 0.07] | [0.17, 0.12] |
| | HD-D | [0.01, 0.01] | [0.03, 0.02] | [0.05, 0.04] | [0.07, 0.06] | [0.09, 0.08] |
| | KL-D | [0.01, 0.01] | [0.01, 0.01] | [0.02, 0.01] | [0.02, 0.01] | [0.02, 0.01] |
| | CPC | [**0.01**, **0.00**] | [**0.01**, **0.00**] | [**0.01**, **0.00**] | [**0.00**, **0.00**] | [**0.00**, **0.00**] |
| M | NJEE | [0.18, 0.20] | [0.18, 0.18] | [0.16, 0.18] | [0.14, 0.17] | [**0.12**, 0.16] |
| | SMILE | [0.08, 0.08] | [0.26, 0.28] | [0.47, 0.48] | [0.66, 0.61] | [0.90, 0.80] |
| | GAN-D | [0.03, 0.02] | [0.05, 0.04] | [0.09, 0.05] | [**0.13**, **0.08**] | [0.18, **0.15**] |
| | HD-D | [0.02, 0.01] | [0.05, 0.04] | [0.11, 0.08] | [0.21, 0.15] | [0.31, 0.26] |
| | KL-D | [**0.01**, **0.01**] | [**0.03**, **0.02**] | [**0.06**, **0.04**] | [0.17, 0.11] | [0.49, 0.33] |
| | CPC | [0.01, 0.01] | [0.13, 0.06] | [1.16, 0.64] | [5.38, 3.48] | [15.67, 11.38] |

Table 4: Bias (B), variance (V), and MSE (M) of the MI estimators using the joint architecture, when $d = [5, 10]$ and $N = 64$, for the Gaussian setting. Each $f$-DIME estimator is abbreviated to $f$-D.

| | | Gaussian | | | | |
|---|---|---|---|---|---|---|
| | MI | 2 | 4 | 6 | 8 | 10 |
| B | NJEE | [0.30, 0.29] | [**0.03**, 0.13] | [0.46, **0.06**] | [1.23, 0.38] | [2.35, 0.80] |
| | SMILE | [0.29, 0.24] | [0.61, 0.52] | [0.76, 0.68] | [0.85, 0.71] | [0.96, 0.68] |
| | GAN-D | [0.06, 0.12] | [0.09, 0.17] | [**0.14**, 0.17] | [**0.06, 0.20**] | [**0.30, 0.18**] |
| | HD-D | [0.04, 0.09] | [0.09, 0.14] | [0.15, 0.22] | [0.28, 0.39] | [0.53, 0.40] |
| | KL-D | [**0.04, 0.07**] | [0.09, **0.13**] | [0.19, 0.30] | [0.40, 0.58] | [0.93, 1.05] |
| | CPC | [0.17, 0.20] | [0.80, 0.89] | [2.10, 2.20] | [3.89, 3.93] | [5.85, 5.86] |
| V | NJEE | [0.04, 0.05] | [0.06, 0.08] | [0.09, 0.10] | [0.15, 0.13] | [0.27, 0.13] |
| | SMILE | [0.06, 0.06] | [0.09, 0.13] | [0.12, 0.20] | [0.23, 0.32] | [0.46, 0.46] |
| | GAN-D | [0.05, 0.06] | [0.08, 0.12] | [0.13, 0.19] | [0.24, 0.30] | [0.69, 0.52] |
| | HD-D | [0.05, 0.06] | [0.08, 0.11] | [0.12, 0.16] | [0.20, 0.24] | [0.57, 0.49] |
| | KL-D | [0.04, 0.05] | [0.06, 0.08] | [0.06, 0.10] | [0.06, 0.10] | [0.06, 0.10] |
| | CPC | [**0.03, 0.04**] | [**0.02, 0.03**] | [**0.01, 0.01**] | [**0.00, 0.00**] | [**0.00, 0.00**] |
| M | NJEE | [0.13, 0.13] | [**0.06, 0.09**] | [0.30, **0.10**] | [1.66, **0.28**] | [5.78, 0.76] |
| | SMILE | [0.14, 0.11] | [0.46, 0.40] | [0.70, 0.66] | [0.95, 0.83] | [1.37, 0.93] |
| | GAN-D | [0.06, 0.08] | [0.09, 0.15] | [0.15, 0.22] | [0.24, 0.34] | [**0.78, 0.55**] |
| | HD-D | [0.05, 0.07] | [0.09, 0.13] | [0.15, 0.21] | [0.28, 0.40] | [0.86, 0.65] |
| | KL-D | [**0.04, 0.06**] | [0.07, 0.10] | [**0.10**, 0.19] | [**0.22**, 0.44] | [0.92, 1.20] |
| | CPC | [0.06, 0.08] | [0.67, 0.83] | [4.42, 4.84] | [15.14, 15.45] | [34.22, 34.32] |

Table 5: Bias (B), variance (V), and MSE (M) of the MI estimators using the joint architecture, when $d = 20$ and $N = 64$, for the Gaussian setting. Each $f$-DIME estimator is abbreviated to $f$-D.

| | | Gaussian | | | | |
|---|---|---|---|---|---|---|
| | MI | 2 | 4 | 6 | 8 | 10 |
| B | NJEE | 0.29 | **0.18** | **0.01** | **0.17** | 0.37 |
| | SMILE | 0.18 | 0.37 | 0.44 | 0.50 | 0.52 |
| | GAN-D | 0.17 | 0.27 | 0.35 | 0.34 | **0.26** |
| | HD-D | 0.16 | 0.28 | 0.43 | 0.61 | 0.73 |
| | KL-D | **0.13** | 0.25 | 0.48 | 0.87 | 1.44 |
| | CPC | 0.25 | 0.98 | 2.29 | 3.99 | 5.88 |
| V | NJEE | 0.06 | 0.10 | 0.13 | 0.17 | 0.16 |
| | SMILE | 0.05 | 0.11 | 0.18 | 0.30 | 0.51 |
| | GAN-D | 0.06 | 0.11 | 0.19 | 0.32 | 0.55 |
| | HD-D | 0.06 | 0.11 | 0.20 | 0.29 | 0.43 |
| | KL-D | 0.05 | 0.09 | 0.11 | 0.12 | 0.11 |
| | CPC | **0.04** | **0.03** | **0.01** | **0.00** | **0.00** |
| M | NJEE | 0.14 | **0.14** | **0.13** | **0.20** | **0.30** |
| | SMILE | 0.09 | 0.25 | 0.38 | 0.55 | 0.79 |
| | GAN-D | 0.09 | 0.19 | 0.31 | 0.43 | 0.62 |
| | HD-D | 0.09 | 0.19 | 0.39 | 0.66 | 0.96 |
| | KL-D | **0.07** | 0.15 | 0.34 | 0.87 | 2.19 |
| | CPC | 0.10 | 0.99 | 5.25 | 15.89 | 34.57 |

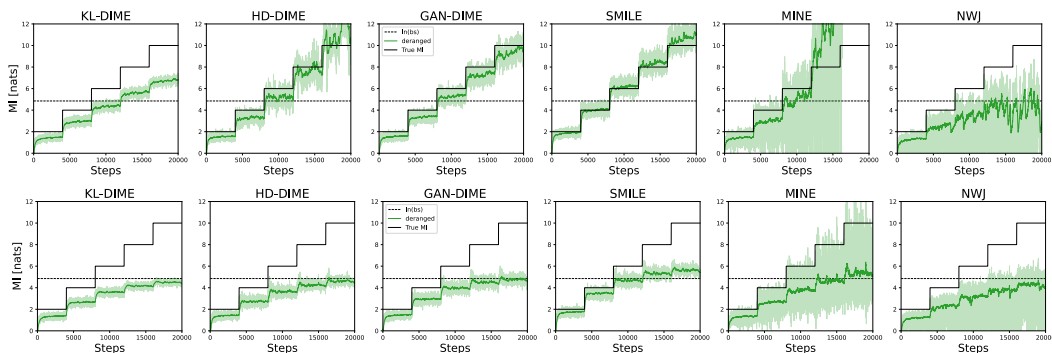

Figure 16: MI estimates when $d = 20$ and $N = 128$, top row: derangement strategy; bottom row: permutation strategy.

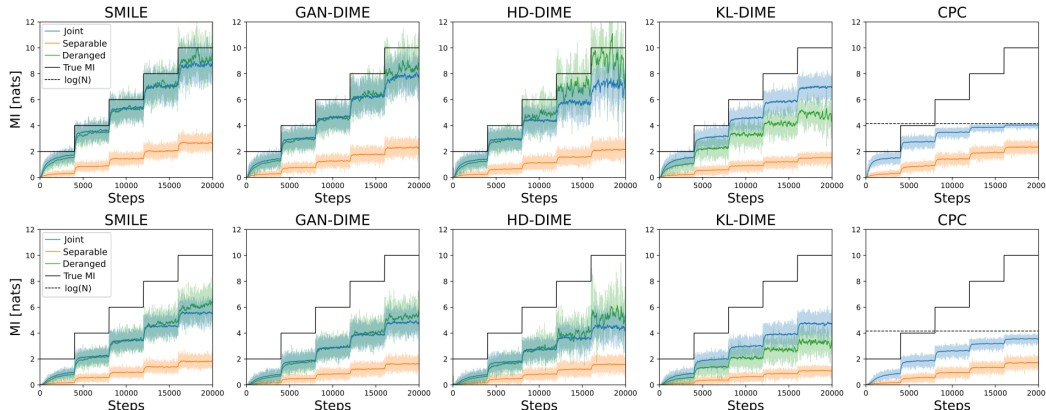

Figure 17: MI estimates when $d = 100$ and $N = 64$. The *Gaussian* setting is represented in the top row, while the *cubic* setting is shown in the bottom row.

The class $f$-DIME is able to estimate the MI for high-dimensional distributions, as shown in Fig. 17, where $d = 100$. In that figure, the estimates behavior is obtained by using the simple architectures described in Sec. D.1 of the Appendix. Thus, the input size of these neural networks (200) is comparable with the number of neurons in the hidden layers (256). Therefore, the estimates could be improved by increasing the number of hidden layers and neurons per layer. The graphs in Fig. 17 illustrate the advantage of the architecture *deranged* over the *separable* one.

### D.1.3 Considerations on Derangements

To facilitate the understanding of the role of derangements during training, we provide a practical example in the following.

Suppose for simplicity that $N = 3$. Then, a random permutation of $\mathbf{y} = [y_1, y_2, y_3]$ can be $[y_2, y_3, y_1]$, where the number of fixed points is $K = 0$ as no elements remain in the same position after the permutation. However, another permutation of $\mathbf{y}$ is $[y_1, y_3, y_2]$. In this case, it is evident that $y_1$ remains in the same initial position, and the number of fixed points is $K = 1$. A random derangement of $\mathbf{y} = [y_1, y_2, y_3]$, instead, ensures by definition that no element of $\mathbf{y}$ ends up in the same initial position, contrarily from a naive random permutation. This idea is essential to avoid having shuffled marginal samples that actually are realizations of the joint distribution. In fact, we proved that a random permutation strategy would lead to a biased estimator (see the permutation bound in Corollary 5.3).

It is extremely important to remark that the derangement sampling strategy it is not only applicable to $f$-divergence based estimators, rather, any discriminative variational estimator should use it to avoid upper bound MI estimates, as it can be observed from Fig. 16

### D.1.4 Summary of the Estimators

We give an insight on how to choose the best estimator in Tab. 6, depending on the desired specifics. We assign qualitative grades to each estimator over different performance indicators. All the indicators names are self-explanatory, except from *scalability*, which describes the capability of the estimator to obtain precise estimates when $d$ and $N$ vary. The grades ranking is, from highest to lowest: ✓✓, ✓, $\sim$, ✗. When more than one architecture is available for a specific estimator, the grade is assigned by considering the best architecture within that particular case.

Even though the estimator choice could depend on the specific problem, we consider $I_{GAN-DIME}$ to be the best one. The rationale behind this decision is that $I_{GAN-DIME}$ achieves the best performance for almost all the indicators and lacks weaknesses. Differently, $I_{CPC}$ estimate is upper bounded, $I_{SMILE}$ achieves slightly higher bias, and $I_{NJEE}$ is strongly $d$ and $N$ dependent. However, if the considered problem requires the estimation of a low-valued MI, $I_{KL-DIME}$ is slightly more accurate than $I_{GAN-DIME}$.

One limitation of this paper is that the set of $f$-divergences analyzed is restricted to three elements. Thus, probably there exists a more effective $f$-divergence which is not analyzed in this paper. Another limitation is that $f$-DIME does not result in neither a lower nor an upper bound on the true MI. Nonetheless, in the following subsection, we show that it is actually possible to obtain a VLB version of the estimator.

### D.1.5 Lower Bound Adaptation

As discussed above, the $f$-DIME estimator does not constitute a lower bound on the true MI. This is due to two main reasons that make $f$-DIME different from the others VLB MI estimators:

1. The general value function $\mathcal{J}_f$ in (5) is the dual representation of the more general $f$-divergence and the KL-divergence is only one special case. Notice that the value $\mathcal{J}_f$ is a lower bound on the $f$-divergence;

2. We exploit the maximizer of $\mathcal{J}_f$, i.e. $\hat{T}$, to build the MI estimator at inference time. This is a key component that allows us to get rid of the partition function for MI estimation, and it comes at the expense of not having a lower bound estimator.

However, and perhaps quite remarkably, $f$-DIME can be adjusted to be a lower bound on the MI by adding the partition term (in the SMILE, MINE or NWJ fashion) during inference time. One way to do such adaptation is to use the extracted density ratio inside the expressions of NWJ or MINE, as in the following:

$$I_{fDIME_{NWJ}}(X;Y) = \mathbb{E}_{(\mathbf{x},\mathbf{y})\sim p_{XY}(\mathbf{x},\mathbf{y})}\left[\log\left(\left(f^*\right)'(\hat{T}(\mathbf{x},\mathbf{y}))\right)\right] \tag{84}$$
$$- \mathbb{E}_{(\mathbf{x},\mathbf{y})\sim p_X(\mathbf{x})p_Y(\mathbf{y})}\left[\left(\left(f^*\right)'(\hat{T}(\mathbf{x},\mathbf{y}))\right)\right] + 1,$$

where $I_{fDIME_{NWJ}}(X;Y)$ is the $f$-DIME estimator obtained using any $f$-divergence dual representation of (5) but with the partition term of the NWJ estimator.

### D.2 Self-Consistency Tests

The benchmark considered for the self-consistency tests is similar to the one applied in prior work [20]. We use the images collected in MNIST [33] and FashionMNIST [34] data sets. Here, we test three properties of MI estimators over images distributions, where the MI is not known, but the estimators consistency can be tested:

1. **Baseline**. $X$ is an image, $Y$ is the same image masked in such a way to show only the top $t$ rows. The value of $\hat{I}(X;Y)$ should be non-decreasing in $t$, and for $t = 0$ the estimate should be equal to 0, since $X$ and $Y$ would be independent. Thus, the ratio $\hat{I}(X;Y)/\hat{I}(X;X)$ should be monotonically increasing, starting from 0 and converging to 1.

2. **Data-processing**. $\bar{X}$ is a pair of identical images, $\bar{Y}$ is a pair containing the same images masked with two different values of $t$. We set $h(Y)$ to be an additional masking of $Y$ of 3

Table 6: Summary of the MI estimators.

| Estimator | Low MI | | High MI | | Scalability |
|---|---|---|---|---|---|
| | **Bias** | **Variance** | **Bias** | **Variance** | |
| $I_{KL-DIME}$ | ✓✓ | ✓✓ | ~ | ✓✓ | ✓✓ |
| $I_{HD-DIME}$ | ✓✓ | ✓✓ | ✓ | ✓ | ✓✓ |
| $I_{GAN-DIME}$ | ✓✓ | ✓✓ | ✓✓ | ✓ | ✓✓ |
| $I_{SMILE}(\tau=1)$ | ✓ | ✓✓ | ✓ | ✓ | ✓✓ |
| $I_{NJEE}$ | ✓ | ✓✓ | ✓ | ✓✓ | ✗ |
| $I_{CPC}$ | ~ | ✓✓ | ✗ | ✓✓ | ✗ |
| $I_{SMILE}(\tau=\infty)$ | ✓ | ~ | ✓ | ✗ | ✓✓ |
| $I_{MINE}$ | ✓ | ✗ | ✗ | ✗ | ✓✓ |
| $I_{NWJ}$ | ✓ | ✗ | ✗ | ✗ | ✓✓ |

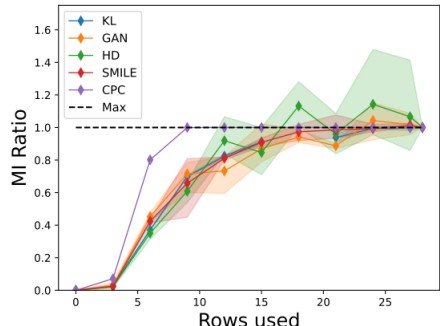

(a) Baseline property, MNIST digits data set.

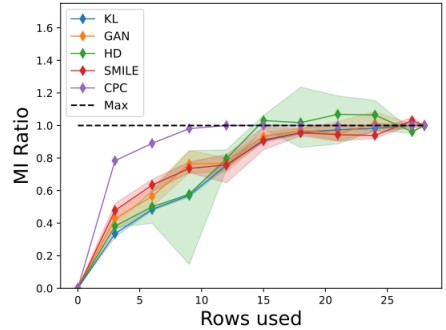

(b) Baseline property, FashionMNIST data set.

Figure 18: Comparison between different estimators for the baseline property, using MNIST data set on the left and FashionMNIST on the right.

      rows. The estimated MI should satisfy $\hat{I}([X,X];[Y,h(Y)])/\hat{I}(X;Y) \approx 1$, since including further processing should not add information.

3. **Additivity**. $\bar{X}$ is a pair of two independent images, $\bar{Y}$ is a pair containing the masked versions (with equal $t$ values) of those images. The estimated MI should satisfy $\hat{I}([X_1,X_2];[Y_1,Y_2])/\hat{I}(X;Y) \approx 2$, since the realizations of the $X$ and $Y$ random variables are drawn independently.

These tests are developed for $I_{fDIME}$, $I_{CPC}$, and $I_{SMILE}$. Differently, $I_{NJEE}$ training is not feasible, since by construction $2d-1$ models should be created, with $d=784$ (the gray-scale image shape is $28 \times 28$ pixels). The neural network architecture used for these tests is referred to as **conv**.

**Conv**. It is composed by two convolutional layers and one fully connected. The first convolutional layer has 64 output channels and convolves the input images with $(5 \times 5)$ kernels, stride $2\,px$ and padding $2\,px$. The second convolutional layer has 128 output channels, kernels of shape $(5 \times 5)$, stride $2\,px$ and padding $2\,px$. The fully connected layer contains 1024 neurons. ReLU activation functions are used in each layer (except from the last one). The input data are concatenated along the channel dimension. We set the batch size equal to 256.

The comparison between the MI estimators for varying values of $t$ is reported in Fig. 18, 19, and 20. The behavior of all the estimators is evaluated for various random seeds. These results highlight that almost all the analyzed estimators satisfy the first two tests ($I_{HD-DIME}$ is slightly unstable), while none of them is capable of fulfilling the additivity criterion. Nevertheless, this does not exclude the existence of an $f$-divergence capable to satisfy all the tests.

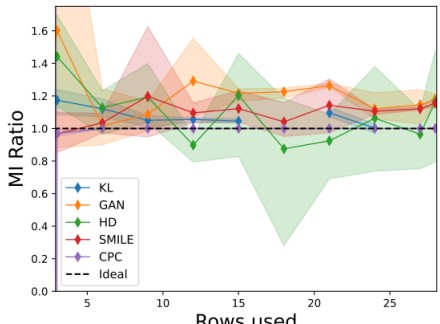 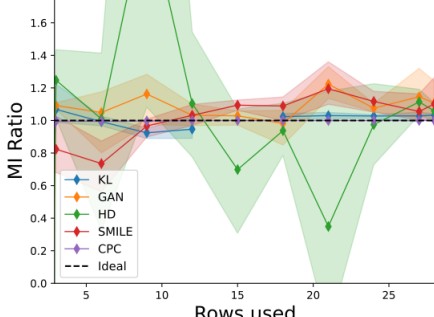

(a) Data processing property, MNIST digits data set.   (b) Data processing property, FashionMNIST data set.

Figure 19: Comparison between different estimators for the data processing property, using MNIST data set on the left and FashionMNIST on the right.

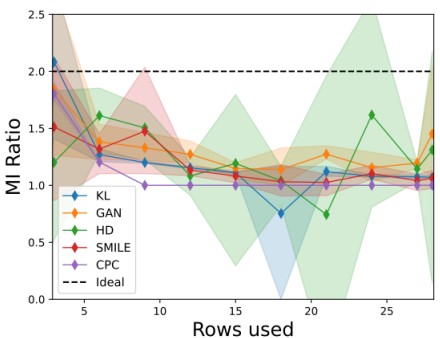 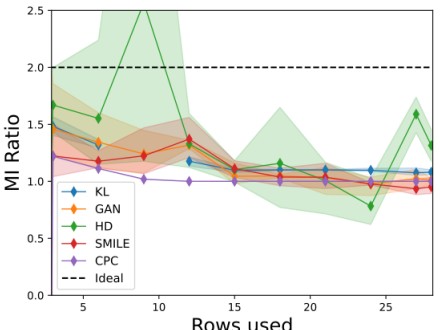

(a) Additivity property, MNIST digits data set.   (b) Additivity property, FashionMNIST data set.

Figure 20: Comparison between different estimators for the additivity property, using MNIST data set on the left and FashionMNIST on the right.

