# OpenReview forum: "Mutual Information Estimation via $f$-Divergence and Data Derangements"
_NeurIPS.cc/2024/Conference — NeurIPS 2024 poster_

### Official Review · Reviewer_mW4u · 2024-07-10

**Soundness:** 4
**Presentation:** 3
**Contribution:** 3
**Rating:** 7
**Confidence:** 3

**Summary:**

This paper introduces a new method for mutual information estimation using a discriminative training approach based on f-divergences. Notably, the authors address a well-known limitation for this class of estimators which exhibit exponential sample complexity in the strong dependence limit. The authors also demonstrate a simple but surprising observation that the method for drawing marginal samples can have a huge impact on estimates, and show how to correctly handle this with "data derangements".

**Strengths:**

- The f-divergence development elegantly avoids partition function estimation.
- While the problem with negative samples is intuitive, it was surprising to see how strong the negative effects could be.
- Between the main text and appendix, a good selection of results studying a wide range of settings. Results look relatively robust.
- It was nice to show a high-dimensional result with images.

**Weaknesses:**

- The theoretical analysis only consider the case when the optimum discriminator is exactly known. It would be nice to have some understanding of the effect of mis-estimation.  Unlike other approaches, we don't get a lower bound on MI, only an estimate.
- Results section could be improved for clarity. I didn't follow the architecture very well, maybe it can be illustrated? (Appendix D helped - maybe it's just a space issue. A schematic in the appendix would still be helpful though.)
- The final conclusions in the Variance Analysis could be stated more clearly.

**Questions:**

Am I correct that one drawback of this procedure is that it doesn't result in an MI lower bound? (Because the objective that is not optimized is not a MI bound, it just guarantees that at the optimum you get a solution that gives the true MI.)  If so, is it easy to understand why other discriminative approaches do give a lower bound and this one doesn't?

Why did the derangement sampling strategy affect time so much? Does it change the convergence speed, or am I missing something else?

**Limitations:**

I think these could be made more explicit.

---

> ### Author Rebuttal · Authors · 2024-08-06
>
> We thank the Reviewer for the positive feedback. We particularly appreciate that the Reviewer caught the key points of our contribution and highlighted the novelty and strengths of our paper. Regarding the questions, please find below our point-to-point response.
>
> **Weaknesses**:
> * We derive in Theorem 3.1 the expression of the proposed class of estimators under the assumptions of optimal convergence. However, as the Reviewer noticed, in practice, due to sub-optimality, it is insightful to know the distance of the MI estimate from the true MI value. For the sake of completeness, we report in the proof of Lemma 3.2 the effect of this discrepancy and show that the convergence of the proposed MI estimator depends also on the choice of the generator function $f$. In particular, from eq. (50) it can be observed that the discrepancy between the true instantaneous MI and the estimated one is controlled also by $f$. Please refer to our answer to Reviewer 3r9p for the detailed formula as no space is left here.
> Therefore, as we stated in the paper (see lines 77, 114, and 246), different divergences have different performance, and this is mostly due to numerical aspects. Overall, we found that GAN-DIME represents the best trade-off.
> * The architecture description in the main text is brief for space limitations and because the joint and separable architectures are already-known in the domain of MI estimation. We will include a schematic in the appendix to improve the clarity (see the one-page pdf). Thanks.
> * In line 148 we write “The great advantage of $f$-DIME is to avoid the partition function estimation step, significantly reducing the variance of the estimator.”, which is the biggest result related to the variance (reinforced in line 152 “Hence, the $f$ -DIME class of models has lower variance than..”). Lemma 4.1 returns a theoretical upper bound on the variance of $f$-DIME, while Lemma 4.2 states that the variance of $f$-DIME is finite when $X$ and $Y$ are correlated Gaussian random variables.  However, we will improve the clarity by separating the paragraph starting with “Furthermore, we provide in Appendix C two supplementary results.”, to emphasize that these two lemmas are additional results w.r.t. the main one previously stated. Thanks.
>
> **Questions**:
>
> * The Reviewer is right about the fact that the $f$-DIME estimator does not constitute a lower bound on the true MI. This is due to two main reasons that make $f$-DIME different from the others VLB MI estimators. 1) The general value function $\mathcal{J_f}$ in eq. (5), utilized during training, is the dual representation of the more general $f$-divergence and the KL-divergence is only one special case. Notice that the value $\mathcal{J_f}$ is a lower bound on the $f$-divergence. 2) We exploit the maximizer of $\mathcal{J_f}(T)$ (i.e., $\hat{T}$) to build the MI estimator, at inference time. This is a key component that allows us to get rid of the partition function for MI estimation, and it comes at the expense of not having a lower bound estimator.
> However, we do not consider it as a significant drawback as from all the numerical experiments provided, the estimates are rather robust and in many cases with lower values compared to overshot estimates obtained with MINE and NWJ which are theoretical lower-bounds. Moreover and perhaps quite remarkably, $f$-DIME can be adjusted to be a lower-bound by adding the partition term (in the SMILE, MINE or NWJ fashion) during inference time. From the code offered by the authors of SMILE in [17] it is clear that SMILE is obtained with such a procedure. We did not discuss how to transform $f$-DIME to a lower-bound estimator as our intention for having low-variance estimate was to get rid-off the partition function, but one way to do such adaptation is to use the extracted density-ratio inside the expressions of NWJ or MINE, as in the following:
>
> $I_{fDIME-{NWJ}}(X;Y) = E_{p_{XY}}\biggl[ \log \biggl(\bigl(f^{*}\bigr)^{\prime}\bigl(\hat{T}(\mathbf{x},\mathbf{y})\bigr) \biggr) \biggr] - E_{p_{X}p_{Y}}\biggl[\biggl(\bigl(f^{\ast}\bigr)^{\prime}\bigl(\hat{T}(\mathbf{x},\mathbf{y})\bigr) \biggr) \biggr]+1$,
>
> where $I_{fDIME-{NWJ}}(X;Y)$ is the $f$-DIME estimator obtained using any $f$-divergence dual representation of eq. (5) but with the partition term of the NWJ estimator.
> We will add this comment in a short dedicated Section D.1.5.
>
> * The derangement procedure does not visibly improve the convergence speed, rather it significantly reduces the amount of samples to be used during each training iteration. In fact, the expectation in the second term of $\mathcal{J_f}$ in eq. (5) only requires $N$ samples obtained with derangements rather than $N(N-1)$ as in the joint architecture (see line 219), thus the deranged architecture has computational complexity $\Omega(N)$. Differently, the complexity of the joint architecture is $\Omega(N^2)$, given that it needs $N(N-1)$ samples every iteration, as we explained in Section 6.1. This time difference is clearly represented in Fig. 8 in the Appendix. Additionally, in the case of i.i.d. samples, the derangement can be developed with a shift-based approach (see line 299). This renders the deranged architecture extremely fast (see Fig. 5).
>
> **Limitations**:
>
> We will include a note in the Appendix Sec. D.1.4 about the fact that $f$-DIME nature renders it neither a lower bound nor an upper bound. We will also add a comment on the interesting trade-off between low-variance estimates and true MI lower-bound in a new section in the Appendix, Sec. D.1.5.
>
> **Rating**:
>
> We hope that our responses clarified the Reviewer’s doubts.
> Given the positive initial feedback, we hope our answers and changes make the contribution of the paper even clearer and stronger.
> If so, please consider increasing the score of the paper, thanks.

---

> > ### Comment · Reviewer_mW4u · 2024-08-09
> > **Response**
> >
> > Thanks for the detailed feedback, and clarification on variance results. I'll raise my score based on that, and partially to balance out the review of dXEo which seems off base.

---

> > > ### Author Response · Authors · 2024-08-09
> > > **We thank the Reviewer**
> > >
> > > We sincerely thank the Reviewer for the positive comment and we are glad to read that the Reviewer increased the score to 7.

---

> ### Comment · Reviewer_mW4u · 2024-08-11
> **Follow-up question on stability / benchmarks**
>
> I just realized that you didn't cite or include the benchmarks from Czyz et al, "Beyond Normal: On the Evaluation of Mutual Information Estimators", NeurIPS 2023. This seems like an omission, though it doesn't directly probe the things you were focusing on.
>
> The reason I realized this is the following. I sent the title of your paper to my students and directed them to see if there is an arxiv version (without telling me the authors of course!). They found it and ran your method on our internal benchmarks, including Czyz et al.
>
> The results were very unstable, and we got NaNs on most of these results. Do you have any thoughts on stability of your results / this benchmark?
> Also, it seems that some of the experiments (self-consistency) were not yet implemented in the code, though I understand cleaning up code often happens closer to final publication.
>
> (I realize this is is quite late in the cycle, and would appreciate even brief high level thoughts /  speculation on these.)

---

> > ### Author Response · Authors · 2024-08-12
> > **Answer on benchmarks and stability**
> >
> > Dear Reviewer,
> >
> > We believe there might be a misunderstanding here.
> > In our paper we reported many tests with the benchmarks from Czyz et al. (we did cite such paper, please check ref. [24]).
> > Figs. 3 and 4 especially describe the excellent results attained by $f$-DIME in those benchmarks.
> > The reason why the Reviewer’s students did not find them in the arxiv version is that such a version is outdated.
> >
> > Regarding the experiments for the Czyz et al. benchmarks, the results we obtained were stable. We noticed that the version of the libraries used for the tests may have an impact. We will specify on GitHub the exact versions of the libraries and code we used to obtain the results in Figs. 3,4. Moreover, we will include a docker image in the repository to facilitate reproduction.
> > The motivation why the code for the self-consistency tests is not provided yet is partially as the Reviewer wrote, and partially because the results of the self-consistency tests are reported in the Appendix, and it is not required to provide code for results in the Appendix. However, we will also include this part of the code on GitHub.
> >
> > Finally, we agree about instability for the self-consistency tests for HD-DIME and some NaNs encountered for KL-DIME (as can be observed from Figs. 17,18,19), while GAN-DIME returned stable results.
> >
> > Thanks.

---

> > > ### Comment · Reviewer_mW4u · 2024-08-12
> > > **Thanks**
> > >
> > > Thanks for all the last-minute clarifications. Sorry I didn’t carefully recheck the results when I went back to investigate about the student results. Thank you for the stability tips, and looking forward to the “camera ready” code.

---

### Official Review · Reviewer_dXEo · 2024-07-10

**Soundness:** 2
**Presentation:** 3
**Contribution:** 3
**Rating:** 4
**Confidence:** 3

**Summary:**

The authors provide a new representation of mutual information (in term of a general $f$-divergence) which has the advantage that the corresponding estimator has a low variance, in stark contrast to the MINE estimator which has a variance which is exponential in the size of the mutual information. The new estimator is tested on multi-dimensional Gaussian distributions.

**Strengths:**

The goal to provide low-variance estimator for mutual information estimation is a timely and very important goal with many potential applications.

**Weaknesses:**

The reviewer is puzzled by the statement of Theorem 3.1 on which the rest of the paper relies.  The issue is the existence of a permutation
$\sigma$ such that $p(\sigma(y)|x) = p(y)$.  Do the author mean to assume the existence of such permutation?  The reviewer may be missing a point but I cannot see why such permutation should exist which makes the permuted random variable $Y$ independent of $X$.  I don't see why this should be true for general data and if it holds true this seems like a very strong assumption because it implies a form independence between $X$ and $Y$.  If this point, which is central to the paper, is not clarified, the paper cannot be considered for publication in the present form.

**Questions:**

Clarify the issue of the existence of the permutation in Theorem 1. Prove its existence, or provide examples to which it applies.

**Limitations:**

The limitations have not been addressed properly, especially in regard about the assumptions of Theorem 3.1.

---

> ### Author Rebuttal · Authors · 2024-08-06
>
> We thank the Reviewer for the time spent reviewing our paper. We understand the Reviewer’s main concern is centered around the permutation law $\sigma(Y)$ which renders $Y$ independent from $X$.
> While it is true that such a function may not exist in practice and we assume its existence in the theorem statement, our contribution is twofold: to study the effect that a random permutation strategy has on MI estimators and propose a novel derangement strategy which solves an important flow derived from random permutations, which have been most often used in the literature.
>
> Please, also notice that we test the proposed family of estimators not only on multi-dimensional Gaussian distributions, but also on both complex Gaussian transformations (half-cube, asinh, and swiss roll mappings, see Fig. 3) and on non-Gaussian distributions (uniform and student distributions, Fig. 4), following the procedure illustrated in [24]. Please find below our detailed answer.
>
> **Weaknesses**:
>
> First of all, a permutation strategy is a basic and quite common technique adopted in most of the neural MI estimators. Its role is to provide a practical way to simulate sampling from the product of marginals $p(x)\cdot p(y)$, and we define $\sigma(\cdot)$ as the permutation function. For instance, in [a], the authors study the maximization of mutual information (for capacity estimation) and use the same notation. Notice that in [a] the authors use as a permutation function a random permutation operation, further demonstrating the importance of our findings on the derangement procedure. In fact, the real issue is that currently many papers (accepted at top ML conferences), like [b] and [c], use a random permutation as practical implementation of such a permutation procedure, which we show in this paper leads to upper bounds. Moreover, in the first neural MI estimator (MINE) the authors also proposed the same random permutation approach [16].
>
> Theorem 3.1 assumes its existence and provides the closed-form expression of the estimator in ideal conditions. However, as also the Reviewer noticed, such a permutation function may not exist in practice and researchers typically adopt a simple random permutation (sometimes also called shuffling mechanism) strategy to break the joint relationships and thus sample from the marginals. We thus study in Sec. 5 the practical implications of such choice and we find out that the naive random permutation strategy actually creates problems, since it renders the MI estimators biased and upper bounded. This is a very important finding as it highlights a severe limitation in existing samplers.
> To overcome such a limitation, we then propose to use a derangement strategy which, by definition, guarantees that no
> sample will end up in the same position when shuffled.
>
> For completeness of explanation, we provide here an example of what we meant with such a definition. Let’s suppose for simplicity that $N=3$. Then, a random permutation of $\mathbf{y} = [y_1, y_2, y_3]$ is $[y_2, y_3, y_1]$, but it can also be $[y_1, y_3, y_2]$. In the latter case, it is evident that $y_1$ remains in the same initial position. Fixed points appear only with permutations. Instead, a random derangement of $\mathbf{y} = [y_1, y_2, y_3]$ ensures that no elements of $\mathbf{y}$ after derangement ends up in the same initial position, differently from a permutation. This observation is crucial for both Lemma 5.1 and Theorem 5.2.
>
> Finally and very importantly, the derangement sampling strategy is generic and can be applied to any MI estimator as illustrated in Fig. 15 in the Appendix.
>
> **References**:
>
> [a] Li, Z., She, R., Fan, P., Peng, C., & Letaief, K. B. (2023). Learning channel capacity with neural mutual information estimator based on message importance measure. IEEE Transactions on Communications.
>
> [b] Hu, Z., Kang, S., Zeng, Q., Huang, K. &amp; Yang, Y.. (2024). InfoNet: Neural Estimation of Mutual Information without Test-Time Optimization. Proceedings of the 41st International Conference on Machine Learning, in Proceedings of Machine Learning Research 235:19283-19303.
>
> [c] Rhodes, B., Xu, K., & Gutmann, M. U. (2020). Telescoping density-ratio estimation. Advances in neural information processing systems, 33, 4905-4916.
>
>
> **Rating**:
>
> We hope that our responses clarified the Reviewer’s doubt expressed in the weaknesses section. We tried to better explain what our contribution is for what concerns the permutation function. If the Reviewer believes the contribution is now clear and evident, please consider increasing the score of the paper, thanks.

---

> > ### Comment · Reviewer_dXEo · 2024-08-13
> >
> > Thank you for the detailed answers.  I will raise the score to take into account better the contributions of the paper on data derangement strategies which are novel and very interesting.  However am still not convinced by the theoretical part of the paper, and it is not correct to state that the authors hav a low variance estimator of the MI.  It may well be that the estimator proposed is a good substitute but the case is theoretically not made.

---

> > > ### Author Response · Authors · 2024-08-13
> > >
> > > We’d like to thank the reviewer for reading our reply and recognizing the novel contribution on data derangement.
> > >
> > > In the second part of the Reviewer’s comment, we believe there still is a misunderstanding about our claim that the proposed MI estimator has low variance and that “the case is theoretically not made”. We politely disagree:
> > >
> > > 1. We explain that $f$-DIME has lower variance w.r.t. the variational lower bound (VLB) estimators (see lines 134, 152, 242 and Eq. (13)).
> > >
> > > 2. This statement is theoretically proved in Section 4. Herein, we begin by summarizing what was found in [17]: the estimation of the partition function is responsible for the high variance condition in VLB MI estimators. Then, we explain that $f$-DIME avoids the partition function estimation, thus decreasing the estimator’s variance. This is shown by Eqs. (12-14).
> > >
> > > 3. Let us try to explain the findings with different terms to ease the understanding. VLB estimators comprise two terms: the first term is the expectation over $p_{XY}$, the second term is the expectation over $p_Xp_Y$. This indeed reflects the notation used, where the variance of VLB estimators is indicated with $Var_{p_{XY}, p_Xp_Y}$. From (11), it is possible to notice that the variance of the VLB estimators is the sum of the variance of two expectations. The variance of the second expectation ($E_{p_Xp_Y}$) is always greater or equal than $0$, which leads to (12). In particular, for high values of true MI, the variance of $f$-DIME ($Var_{p_{XY}}[E_{p_{XY}^M}[\log\hat{R}]]$) is significantly lower than the variance of the VLB estimators, because the second expectation term has exponentially increasing variance with the true MI (demonstrated in [17]).
> > >
> > > 4. Additionally, we reported two Lemmas in the Appendix that provide: 1) an upper bound on the variance, 2) a characterization of the variance when X and Y are correlated Gaussian random variables.
> > >
> > > 5. Finally and for completeness, we briefly discuss the bias/variance comparison between $f$-DIME and the other MI estimators analyzed in the paper, to justify why $f$-DIME attains an excellent bias/variance trade-off. SMILE decreases the variance of VLB estimators by clipping the partition term. Therefore, also SMILE attains variance comparable to $f$-DIME, but it is biased. CPC is a low variance estimator, however it is more biased than all the other estimators considered, since it is upper bounded by $\log(N)$. NJEE is low variance because it uses $2d-1$ neural networks to obtain the prediction. For this reason, NJEE is computationally extremely expensive (please see line 290 and subsequent ones). Furthermore, NJEE is biased for various choices of $d$. In conclusion, $f$-DIME attains lower variance w.r.t. the VLB estimators, and in general exhibits an excellent bias/variance trade-off.
> > >
> > > 6. The experimental results provide further evidence of the findings.
> > >
> > > We hope that this provides more clarity and offers the basis to raise the score further. Thank you again.

---

### Official Review · Reviewer_3r9p · 2024-07-12

**Soundness:** 3
**Presentation:** 3
**Contribution:** 3
**Rating:** 5
**Confidence:** 5

**Summary:**

This paper proposes a novel discriminative mutual information estimator via the form of f-divergence, which exhibits a bias/variance trade-off.  Experimental results show that the proposed estimator is comparable to existing neural estimators.

**Strengths:**

1. The proposed MI estimator avoids directly computing the density ratio and enjoys a bias/variance trade-off.

2. The authors investigate the impact of data derangement and propose a novel training strategy based on data derangement.

3. Theoretical analysis on characterzing the properties of the proposed estimator are provided.

**Weaknesses:**

-	Can the author theoretically show how close the mutual information estimated by using the value function ($J_f(T)$ in Theorem 3.1) is to the true mutual information?
-	The function T(·) in Theorem 3.1 is missing its definition.
-	Eq (10) seems to be a bit of problem, first row: $e^{I(X;Y)} \rightarrow e^{I_{NWJ}^{M,N}}(X;Y)$,
second row: $e^{I(X;Y)} \rightarrow e^{I_{MINE}^{M,N}}(X;Y)$.
-	Figure 1 shows that the proposed estimator still has a large variance like the existing methods. High variance does not seem to be well addressed.
-	The experimental utilizes low-dimensional datasets, which raises concerns about the comprehensive evaluation of the proposed estimator's efficacy in high-dimensional contexts. The authors' investigation does not adequately address the potential implications of high-dimensional data on the estimator's performance.

**Questions:**

Please refer to Weaknesses

---

> ### Author Rebuttal · Authors · 2024-08-06
>
> We thank the Reviewer for the time spent reviewing our paper and for the valuable feedback. The overall opinion and strengths highlighted by the Reviewer are indeed a precise summary of our contribution. We hope that our replies below will help the Reviewer increase his/her good opinion about the paper. Thank you.
>
> **Weaknesses**:
>
> 1. The Reviewer’s question is very relevant, as in practical scenarios, there is always a numerical bias coming from sub-optimality. Lemma 3.2 also studies this effect of not reaching optimality in the MI estimator and quantifies the discrepancy of the instantaneous MI in terms of convergence to the optimal density ratio (or log-density ratio), see (50). In particular, we show that the bias in the instantaneous MI estimate also depends on the generator function of the $f$-divergence as:
>
> $i(X;Y) - \hat{i}_{n,fDIME}(X;Y) \simeq \delta^{(n)} \cdot \Biggl[\frac{\mathrm{d}}{\mathrm{d}T} \log \bigl( \bigl(f^{*}\bigr)^{\prime}(T)\bigr) \biggr \rvert _{T=\hat{T}} \Biggr]$,
>
> where $\delta^{(n)}=\hat{T}-T^{(n)}$ is the displacement between the optimum discriminator $\hat{T}$ and the obtained one $T^{(n)}$ at the iteration $n$, while $\hat{i}_{n,fDIME}(X;Y)$ is the instantaneous MI estimated using $\mathcal{J}_f$ in Theorem 3.1.
>
> 2. Since Theorem. 3.1 is general (not strictly related to neural networks), we will write $T: \mathcal{X}\times \mathcal{Y} \to \mathbb{R}$. Thanks for noticing the lack of domain definition. Nevertheless, in line 102 we specified that “We propose to parametrize $T(x, y)$ with a deep neural network $T_{\theta}$ of parameters $\theta$ and solve with gradient ascent and back-propagation”.
>
> 3. We disagree with the Reviewer’s comment. Eq. (10) is correct as it is written in the paper. As a matter of fact, $I(X;Y)$ represents the true MI, while $I_{NWJ}^{M,N}(X;Y)$ and $I_{MINE}^{M,N}(X;Y)$ are the MI estimators. Eq. (10) is correct because it relates the variance of the estimators with the true MI.
> The Reviewer's misunderstanding may be caused by lack of clarity in the notation. In the paper, we specified that “the variance scales exponentially with the ground-truth MI” and to further improve the clarity, we will substitute "where" in line 142 with “Where $I(X;Y)$ is the true MI and”
>
> 4. Fig. 1 does not show any comparison with existing MI estimators, thus we suppose the Reviewer referred to Fig. 2. In our paper (see line 152) we state that $f$-DIME has lower variance w.r.t. the variational lower bound (VLB) MI estimators (e.g. MINE, NWJ). This can be immediately noticed from Figs. 3-4 but also from Fig.6 and Fig.15 in the Appendix and from Table 1.
> It is true that the variance of $f$-DIME and SMILE, NJEE, CPC are comparable. However, CPC is upper bounded (thus strongly biased), SMILE exhibits overall worse performance than GAN-DIME, and NJEE is biased for various values of $d$ and computationally demanding.
>
> 5. We understand the Reviewer’s comment may be caused by the lack of high-dimensional results in the main document, where we decided to compare our estimator using common benchmarks and scenarios ([24]). However, as other Reviewers have noticed (and we added a sentence in page 7, line 250), we did provide several experiments with higher-dimensional data in the Appendix. In fact, in addition to the plots with dimension $d=20$ in Figs. 9,10, we reported two scenarios of high-dimensional datasets. Fig. 16 shows the challenging case $d=100$, while the section of consistency tests also shows the case of MI estimation of image datasets.
> Lastly, while the estimation of MI for high-dimensional data remains challenging, it is important to remark that we did not claim to have solved such a challenge, but rather to have provided new tools, such as a new family of estimators and a new data sampling strategy, which can empower researchers in their quest.
>
> **Rating**:
>
> We hope that our answers help the Reviewer to clarify some details and improve the understanding of the value of the numerical results. If the Reviewer believes we succeed in doing so, we encourage him/her to increase the rating. Thank you.

---

> > ### Author Response · Authors · 2024-08-13
> > **Approaching the end**
> >
> > Dear Reviewer 3r9p,
> >
> > As the rebuttal period is approaching its end, we would really appreciate your feedback. The other 3 reviewers expressed their satisfaction with our responses, and we would like to know if our answers also clarified your doubts.
> >
> > Thanks again for your valuable time.

---

### Official Review · Reviewer_H3tA · 2024-07-12

**Soundness:** 3
**Presentation:** 3
**Contribution:** 3
**Rating:** 6
**Confidence:** 3

**Summary:**

This paper investigates the long-standing task of estimating mutual information in high-dimensional data. The authors point out that mutual information estimation methods using variational lower bounds suffer from either high bias or high variance, and proposes an alternative solution leveraging the variational representation of the f-divergence. They further claim that the proposed solution, f-divergence mutual information estimators (f-DIME), exhibits an excellent bias/variance trade-off, higher accuracy and lower complexity.

**Strengths:**

1.	The proposed framework is applicable to any f-divergence, generalizing it beyond the commonly-used KL-divergence (Theorem 3.1).
2.	Theoretical analysis on bias-variance trade-off. (1) Bias. The global optimum of the estimator converges to the real value of the mutual information (Lemma 3.2), making it an unbiased estimator when the assumptions are met. (2) Variance. Variance Analysis (section 4) shows that the proposed method equal or lower variance than methods using the variational lower bound.
3.	As shown in Figure 1, the proposed derangement strategy for data sampling enables a lower bias estimation than the permutation strategy, as it is not upper bounded by log(N).
4.	Runtime analysis (Figure 5) is a good addition to the empirical results.

**Weaknesses:**

1.	They benchmark a series of experiments using a wide range of mutual information estimators (Figure 2, 3, 4). However, the empirical results do not seem to support the central claim of “an excellent bias-variance trade-off”. It is clear that CPC is suffering from the log(N) upper bound issue where N is the number of data samples, but the proposed f-DIME method (KL-DIME, HD-DIME, GAN-DIME) do not seem to be consistently better in either bias or variance compared to the competing methods MINE and SMILE.
2.	Minor issue: The methods are not ordered the same way in Figure 2 versus Figure 3 and 4. It would be helpful to order them consistently.
3.	Minor issue: “NWJ” was not defined before first used in line 67.
4. There are non-neural network methods that have good theoretical rates and/or  good experimental results for higher dimensions. They should at least refer to these if not include some comparisons. These include the following:
https://journals.aps.org/pre/abstract/10.1103/PhysRevE.69.066138 (they reference to this but don't compare. This method often performs very well experimentally)
https://doi.org/10.1109/TIT.2021.3100108
https://proceedings.neurips.cc/paper/2015/hash/06138bc5af6023646ede0e1f7c1eac75-Abstract.html
https://www.pnas.org/doi/full/10.1073/pnas.1715593115

**Questions:**

1.	I would ask the authors to address weakness #1.
2.	I believe it would be helpful to quantify the bias and variance in the empirical results (Figure 2, 3, 4), for better comparison across KL-DIME, HD-DIME, GAN-DIME, NJEE, MINE and SMILE. I just noticed that some relevant information is present in Table 1, 2, 3 and 4 of the Appendix. Would the authors consider presenting such results in the main text instead (for instance, as a nested bar plot)?
3.	In the related works section, the authors describe how some mutual information estimators produce high-biased estimates as they are upper bounded by log(N). Could the authors also explain the root cause(s) of high-variance estimators in the same section?

**Limitations:**

Yes, the authors have adequately addressed the limitations and potential negative societal impact of their work.

---

> ### Author Rebuttal · Authors · 2024-08-06
>
> We thank the Reviewer for the time spent reviewing our paper. In particular, we thank the Reviewer for having spotted the strengths of the paper, it is clear from her/his review that the key points and contributions of the paper have been captured. We provide below our answer to the detailed comments.
>
> **Weaknesses**:
>
> 1. While it is true that we benchmarked $f$-DIME with several other estimators and that the choice of the generator function $f$ does impact the performance of the estimator, the GAN-DIME estimator (proposed as the best $f$-DIME estimator in lines 282 and 1029) actually consistently showed the best performance in all the experiments, especially over MINE and SMILE.
> As a matter of fact, Fig. 2 for SMILE and Figs. 3 for both MINE and SMILE illustrate how they have higher bias compared to GAN-DIME. Moreover, MINE also exhibits large variance w.r.t. to any $f$-DIME estimator. The high-variance phenomenon (theoretically discussed in Sec. 4, eq. 10) of MINE is noticeable for high values of true MI. The reason why this phenomenon is not evident in Fig. 4, is because the plots are obtained for small values of true MI. Nevertheless, the bias of the two estimators is visible also there.
> We did not report MINE in all the experiments as its performance was consistently worse than GAN-DIME.
> Additional results on the high-variance of MINE, however, are shown in Fig. 6 and Fig. 15 in the Appendix, while numerical values are reported in Table 1.
> SMILE, which is known to perform better than MINE (see [17]), performs worse than GAN-DIME in the scenarios illustrated in Fig. 2-4, which are the state-of-the-art benchmarks ([24]). This is even clearer (for many values of true MI) in the Gaussian case of Fig. 2, the half-cube and swiss roll cases of Fig. 3 and the scenarios of Fig. 4.
>
> 2. We thank the Reviewer for noticing that, we will sort them accordingly.
>
> 3. We will rewrite the sentence in line 65 as: “Another variational lower bound based on the KL divergence dual representation introduced in [15] leads to the NWJ estimator (also referred to as f-MINE in [16])”
>
> 4. We thank the Reviewer for mentioning relevant related work. As we have explained in both the abstract and the introduction, to contextualize our contribution and facilitate the analysis, we compare $f$-DIME with state-of-the-art neural estimators. While we also agree that non-neural methods have great performance in certain low-dimension scenarios, it is widely known that they fail to scale at larger dimensions (see [a]). For instance, KSG fails to obtain reasonable estimates even beyond 5 dimensions [b]. Results in [16] demonstrate the superiority of MINE wrt KSG for 20-dimensional multivariate Gaussians.
> Moreover, none of the mentioned papers offer experiments with dimension $d$ larger than 10. Interestingly, the independence test in [c] utilized random permutations instead of derangements. We will include [c] and [d] in the related work.
>
>
> **Questions**:
> 1. See above.
>
> 2. We are glad to read that the Reviewer checked the Appendix carefully and we agree with the importance of such tables. We will include Tab. 1 as a nested bar plot in the main if the paper gets accepted (see the rebuttal pdf), as we will have an additional page of content to fill in. We thank the Reviewer for suggesting the nested bar plot.
>
> 3. Due to limited space constraints, we wrote “the partition function does not need to be computed, leading to low variance estimators”. Implicitly, this sentence reveals that the cause of high variance in MI estimators resides in the partition function estimate. We understand it is a subtle concept for new researchers entering the domain, therefore we will insert in the camera ready version the sentence “the SMILE estimator was introduced in [17], where the authors proved that the estimate of the partition function is the cause for high-variance in variational lower bound estimators.”
>
> **References**
>
> [a] Gao, Shuyang, Greg Ver Steeg, and Aram Galstyan. "Efficient estimation of mutual information for strongly dependent variables." Artificial intelligence and statistics. PMLR, 2015.
>
> [b] Mukherjee, S., Asnani, H., & Kannan, S. (2020, August). CCMI: Classifier based conditional mutual information estimation. In Uncertainty in artificial intelligence (pp. 1083-1093). PMLR.
>
> [c] Zeng, Xianli et al. “Jackknife approach to the estimation of mutual information.” Proceedings of the National Academy of Sciences 115 (2018): 9956 - 9961.
>
> [d] Kevin R. Moon, Kumar Sricharan, and Alfred O. Hero. 2021. Ensemble Estimation of Generalized Mutual Information With Applications to Genomics. IEEE Trans. Inf. Theor. 67, 9 (Sept. 2021), 5963–5996.
>
> **Rating**:
>
> We appreciate the Reviewer recognizing the theoretical and numerical contributions. We hope that our answers and changes clarify the Reviewer’s doubts and can contribute to grading our paper more positively. Thank you.

---

> > ### Comment · Reviewer_H3tA · 2024-08-13
> > **Satisfactory answers.**
> >
> > The answers are satisfactory to me. I appreciate the clarification of 1 and this should be written more emphatically in the paper. I will raise my score to a 6.

---

> > > ### Author Response · Authors · 2024-08-13
> > >
> > > We deeply thank the Reviewer for the positive comment and we will indeed write the clarification of 1 more emphatically in the paper. To ensure accurate reflection of the Reviewer's comment, we kindly request an update of the score to 6 before the rebuttal period ends, as we still do not see this change reflected. Thank you.

---

### Author Rebuttal · Authors · 2024-08-06

We thank the Reviewers for their valuable feedbacks.

Please find below the one-page PDF containing extra information supporting our point-to-point rebuttal.

Sincerely,


The Authors

---

### Decision · Program_Chairs · 2024-09-25

**Decision:**

Accept (poster)

**Comment:**

The authors propose a mutual information estimator using data derangements and explicitly takes permutation invariance into account. Reviewers appreciated the results and how it resolved the exponential complexity of this MI estimation, although there are concerns with the clarity of the paper. Authors, please incorporate the suggestions by the reviewers and make every effort to clarify the writing and presentation.